



# Comparing Abnormalities in Onshore and Offshore Vertical Wind Profiles

Mathias Møller[1], Piotr Domagalski[2,3], and Lars Roar Sætran[1]

[1]Department of Energy and Process Engineering, Norwegian University of Science and Technology, 7491 Trondheim, Norway
[2]Institute of Turbomachinery, Lodz University of Technology, Wolczanska 219/223, 90-924, Lodz, Poland
[3]WindTak LLC, Wroblewskiego street 38a, 93-578 Lodz, Poland

**Correspondence:** Mathias Møller (mathias.moller1@gmail.com), Piotr Domagalski (piotr.domagalski@windtak.pl)

**Abstract.** Understanding the vertical wind profile is paramount for design & operation of wind turbines. It is needed not only for extrapolation of the wind velocity to hub height but also for structural load calculations, to name the most obvious issues. As wind turbines grow in size and development transitions offshore, issues such as shallow surface layers, low-level jets and internal boundary layers are raising questions to the applicability of the commonly used Monin-Obukhov similarity theory to accurately describe the vertical wind development to modern wind turbine hub heights. In this study the 10-minute averaged vertical wind profile up to a minimum elevation of 100m is analyzed through measurements collected from seven sites which represent a span of conditions. Three sites are located offshore in the North/Baltic Sea with varying fetch, two onshore by the Norwegian coast, one further onshore by the Danish coast, and one is an inland forested site in Sweden. Through analysis of data series ranging from 8 months to several years depending on the site, the wind profile has been quantitatively categorized according to the number of exhibited local maxima which are not possible within Monin-Obukhov similarity theory. The results reveal that the occurrence of local maxima scales inversely to the roughness length, causing $65-75\%$ abnormal profiles offshore which decreases as the location transitions from offshore to coastal to further inland, and is lowest at the forested site. The results indicate that issues in predicting the vertical wind profile are most prevalent offshore, where very stable inflections cause severe deviations which may be related to an offshore internal boundary layer. These findings suggest that there is evident need of an improved vertical wind profile description in order to improve the accuracy of power predictions and load calculations, especially at offshore and coastal sites.

## 1 Introduction

As the extent of wind energy extraction grows, there is and has been an increasing focus on wind energy at offshore locations (Nunalee and Basu, 2014). In 2018 wind energy accounted for 48% of total installed power capacity in the EU, the most of any power generation (WindEurope, 2019). 26% of this capacity was installed offshore, down 16% from the record year 2017. When deciding where to build and commission wind farms, knowing the wind speed which crosses the wind turbine area is crucial in assessing the site feasibility. Advancement in technology is enabling such measurements to be performed by Lidar devices which can extend their measuring range to typical turbine hub height elevations. However, when the rotor disc wind



speed is not measured, the assessment relies on models to extrapolate the wind speed to the relevant elevations (Sempreviva et al., 2009). These models may also be used when extracting wind speeds retained from numerical weather prediction tools, or through predictive energy yield calculations. The accuracy of the method for extrapolating the wind speed is evidently crucial, and relies on an understanding of the underlying physics causing the wind speed development. A correct vertical wind profile

(VWP) description is also important in power predictions at operational wind farms which lessens the need of short term energy storage and increases the park efficiency (Kalvig et al., 2014). Additionally, the wind speed and the wind shear are important when assessing turbine loads (Eggers Jr. et al., 2003).

The vertical development of velocity in the surface layer of the atmosphere may be theoretically described through the framework of Monin-Obukhov similarity theory (MO-theory, MOST) (Arya, 1988). MO-theory assumes constant vertical

fluctuations of temperature, velocity and shear stress, sufficient time averaging and a uniform surface roughness, (Foken, 2017). Under these assumptions MOST enables the description of the velocity development with height $u(z)$ through the logarithmic law (Eq. 1).

$$u(z) = \frac{u_*}{k} \left[ ln \left( \frac{z}{z_0} \right) - \psi \left( \frac{z}{L} \right) \right] \tag{1}$$

In Eq. 1 L is the *Obukhov length* which describes the relative importance of buoyant and mechanical effects in atmospheric

turbulence, $z_0$ is the roughness length, $k$ is the von Kármán constant, and $u_*$ is the *friction velocity* (Stull, 2017). The value of $\psi$ changes with atmospheric stability and is negative during stable atmospheric conditions, zero for the special neutral case, and positive in unstable conditions. The determination of the stability function $\psi$ must be done empirically which was a large focus after the theory was initially presented (Foken, 2006). The Kansas field experiment of 1968 largely validated Monin-Obukhov theory as accurately describing the vertical wind profile within the surface layer over flat homogeneous terrain using a 32m

high mast (Kaimal and Wyngaard, 1990).

Another commonly used vertical wind profile description is the empirically proven power law (Eq. 2) which is due to its relative simplicity commonly employed in turbine engineering (Emeis, 2013).

$$u(z) = u(z_r) \left( \frac{z}{z_r} \right)^{\alpha} \tag{2}$$

In Eq. 2 $z_r$ is a reference height where the wind velocity $u(z_r)$ has been measured. The power coefficient $\alpha$ has traditionally

been assumed constant over the vertical extrapolation range, but if applied over ranges exceeding 10-20m should be described as a function of height and atmospheric stability (Emeis, 2014). Although the simplicity of the power law in its original form with $\alpha = const$ makes its use tempting, the lack of connection with the underlying physics makes it less relevant in atmospheric boundary layer (ABL) research.

MO-theory has for a long time been a commonly applied theoretical framework of describing the surface layer winds

relevant for wind turbine engineering, studies are however revealing that the height limitations of its applicability may make it less suitable for common turbine heights (Gualtieri, 2019). Onshore Gryning et al. (2007) found progressive deviations from





the scaling predicted by MO-theory above 50-80m in a study of wind over flat and homogeneous terrain. The study proposed additional length scalings enabling the description of the vertical wind profile through the entire atmospheric boundary layer (ABL) which better replicated measured values. In a review of issues in wind energy meteorology Emeis (2014) also highlights the importance of implementing a unified vertical wind profile description which is not solely valid in the surface layer. The

limitations of Monin-Obukhov theory are known to become more pronounced under stable stratification of the atmosphere when buoyant forces are negative, which often causes shallow surface layers (Emeis, 2013).

Onshore winds in the vicinity of obstacles and surface changes are also known to be prone to internal boundary layer formations which cause deviations in the vertical wind profile. An IBL due solely to a change in surface roughness has been found through measurements at the onshore site Cabauw (Verkaik and Holtslag, 2007), while a combination of a step change

in both surface roughness and temperature was found to cause an IBL development at the onshore site Høvsøre located less than 2km from the sea (Peña et al., 2016). The onshore internal boundary layer is however a short-lived phenomenon due to the increased mixing caused by high surface roughness.

While the aforementioned studies describe some of the issues found in onshore environments, the low surface roughness and large heating capacity of the sea makes the offshore use of MO-theory complicated. Lange et al. (2004) studied the vertical

wind profile 11km offshore in the Danish Baltic Sea and found that MO-theory systematically under-predicted the wind speed at 50m during near-neutral and stable conditions. Tambke et al. (2005) also found larger than predicted wind speeds at 62m at Horns Rev in the North sea located 18km offshore, which was observed for all stability conditions. There is however ambiguity in these findings, as Peña et al. (2008) found an opposite result, namely that surface-layer theory *over-predicted* the wind speed at elevations above 30-40m during stable atmospheric conditions. Implementing the Gryning et al. (2007) correction provided

better agreement with the measured wind speeds. These findings were supported by Sathe et al. (2012) who also found an over-prediction of the wind speed by surface-layer theory at higher altitudes during stable atmospheric conditions which could be accounted for by employing the Gryning et al. (2007) correction.

An explanation of these incorrectly predicted offshore wind speeds during stable conditions may be emerging in the form of a stable offshore internal boundary layer. The offshore stable internal boundary layer is associated with a change in both

surface roughness and temperature and its evolution has been described by Csanady (1974) and Smedman et al. (1997). When warm air on land transitions offshore to a colder sea, an internal boundary layer develops where the air is cooled from the sea. The lower air will after a distance approach the sea temperature, while a very stable inversion lid has developed above. Lange et al. (2004) suggests this inversion lid may be categorized by larger than expected wind speed gradients which were not well predicted by MOST, but could be partly accounted for through an inversion height correction. While most studies on the stable

offshore IBL have been performed in the Baltic Sea, the limited heat flux through this inversion lid means that coastal effects may persist for several hundred kilometers offshore before the temperature differences dissolve, and signs of a distinct thermal layering have been found in the North Sea 80km offshore at the FINO3 research mast (Argyle and Watson, 2014).

While the issue of incorrectly predicting the vertical wind profile has mainly been associated with stable stratification at offshore sites, Riedel et al. (2005) suggests that at the FINO1 site located 45km offshore, the vertical wind gradient was

over-predicted during unstable conditions and under-predicted during stable conditions. Other studies have however found that





MOST is satisfactory in correctly predicting wind shear at offshore locations (Peña et al. (2008), Sathe et al. (2012), Argyle and Watson (2014)).

The deviations between vertical wind profile models and measurements at higher altitudes during stable stratification may be coupled to low-level jets which are known to cause deviations between wind speed measurements and models (Svensson
et al., 2016). The main focus of research has previously been on the onshore nocturnal LLJ which may occur at typical turbine hub heights of 100-200m AGL (Nunalee and Basu, 2014). The offshore low-level jet lacks the same level of understanding, but offshore low-level jets in the Baltic Sea have been found analogous to the onshore nocturnal low level jet, which reached elevations as low as 30-150m (Smedman et al., 1995).

Evidently the limitations of MO-theory to only being applicable within one layer of uniform vertical fluctuations in the
atmospheric boundary layer makes its use limited for wind energy applications where internal boundary layers and shallow surface layers prevail. The identification of these phenomenon is however not simple, a growing body of methods are therefore emerging for assessing deviations from the common vertical wind profile formulations. In a study of the offshore vertical wind profile at FINO1, Kettle (2014) simply categorized the VWP as abnormal if it exhibited a local maximum and thus did not conform with the monotonically increasing behaviour predicted by MO-theory, Most of the profiles were in fact found to
exhibit one or more local maxima, and even cases of the wind monotonically decreasing with height were identified. Local maxima in the vertical wind profile were also discovered by Wagner et al. (2009), who found the negative shear above the maximum to have a large impact on available power when accounting for wind shear across the rotor diameter.

Maxima or 'kinks' in the vertical wind profile may be used both onshore and offshore to describe the height of a surface layer discontinuity (Garratt, 1990). In the present study large datasets primarily comprising several years of 10-minute averaged
vertical wind profile measurements will be analyzed for the occurrence of abnormalities in the form of local maxima. The profiles are measured at 7 locations in onshore, offshore, coastal and forested environments from near-surface elevations up to a height of 100-140m depending on the site. The method of identifying local maxima is chosen due to its simplicity while additionally having a natural coupling with phenomenon associated with discontinuities in the atmospheric boundary layer. The goal is to map how these abnormalities occur and change with site location characteristics, and understand how they are
correlated to atmospheric features such as wind speed and stability. The findings are also assessed in terms of the possibility of these profiles causing significant deviations in the common vertical wind profile descriptions. Based on this the need for more accurate vertical wind profile descriptions at both onshore and offshore sites can be assessed.

## 2 Method

### 2.1 Abnormal profile identification

In the process of identifying abnormal vertical wind profiles, the method of identifying local maxima previously implemented by Kettle (2014) is employed. In this method a 10-minute averaged profile is categorized as abnormal if the velocity profile is not monotonically increasing for all heights, and the abnormal profiles can subsequently be categorized by the number of inflections they exhibit. This method was chosen due to the robustness in that all profiles can be placed with certainty within

one category, while simultaneously enabling the identification of discontinuities in the layering of the atmospheric boundary layer which may be associated with kinks in the velocity profile (Garratt, 1990). The time-averaging of the profiles was not extended to longer periods that the 10-minute average since the project aims at describing dynamical discontinuities which are simultaneously within the range of where classical MO-theory becomes applicable (Petersen et al., 1998).

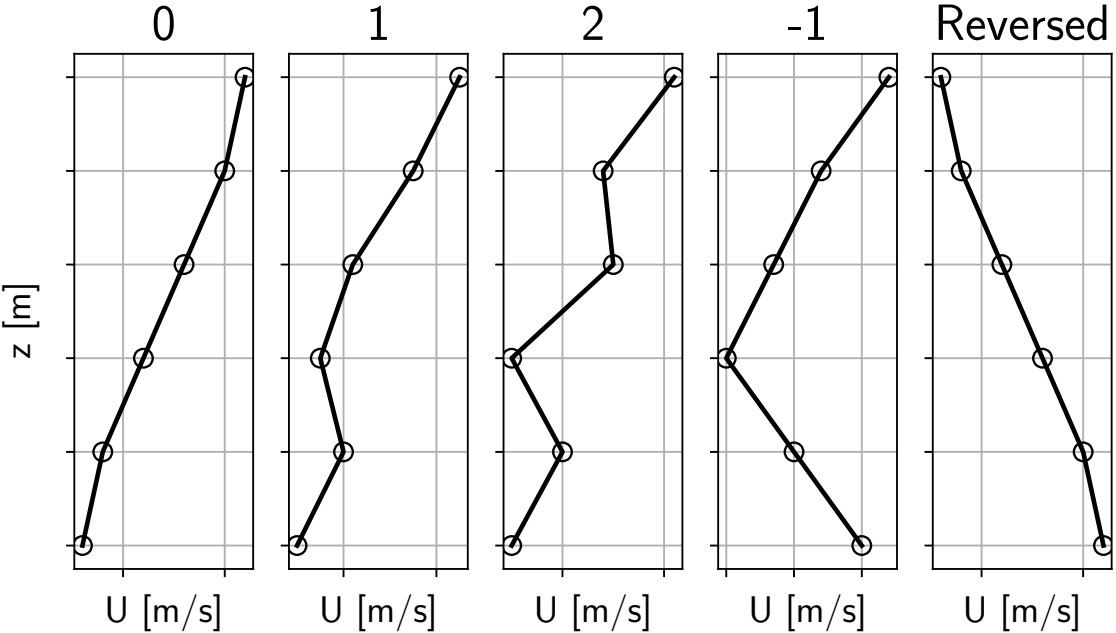

**Figure 1.** Possible vertical wind profile categories when categorizing according to the number of local maximum at a site with 6 measurement heights. Titled according to the number of maxima.

5     The number of profile maximum possible is a function of the number of measurement heights, and can be described as $N_{max} = floor\left(\frac{N_h - 1}{2}\right) = \lfloor\frac{N_h - 1}{2}\rfloor$ where $N_h$ is the number of measurement heights, and $N_{max}$ is the highest number of maximum possible in the vertical wind profile. For a site with 6 measurement sites this would allow at most 2 profile maximum. In addition, a profile with 0 local maximum may exhibit 1 local miminum where the velocity profile is decreasing up to this height and thereafter increasing. This category was appropriately named the -1 local maxima category, or the 1 minimum cate-

10 gory. Profiles were also found where the velocity development was reversed and monotonically decreasing at all measurement heights. An example of the possible profiles for a site with six measurement heights is shown in Fig. 1.

## 3   Data description

Measurements from a total of seven sites were studied in this analysis, starting at heights between 10-40m, and extending to 100-140m depending on the site instrumentation. The number of height measurements varied from 6-8 except for one site



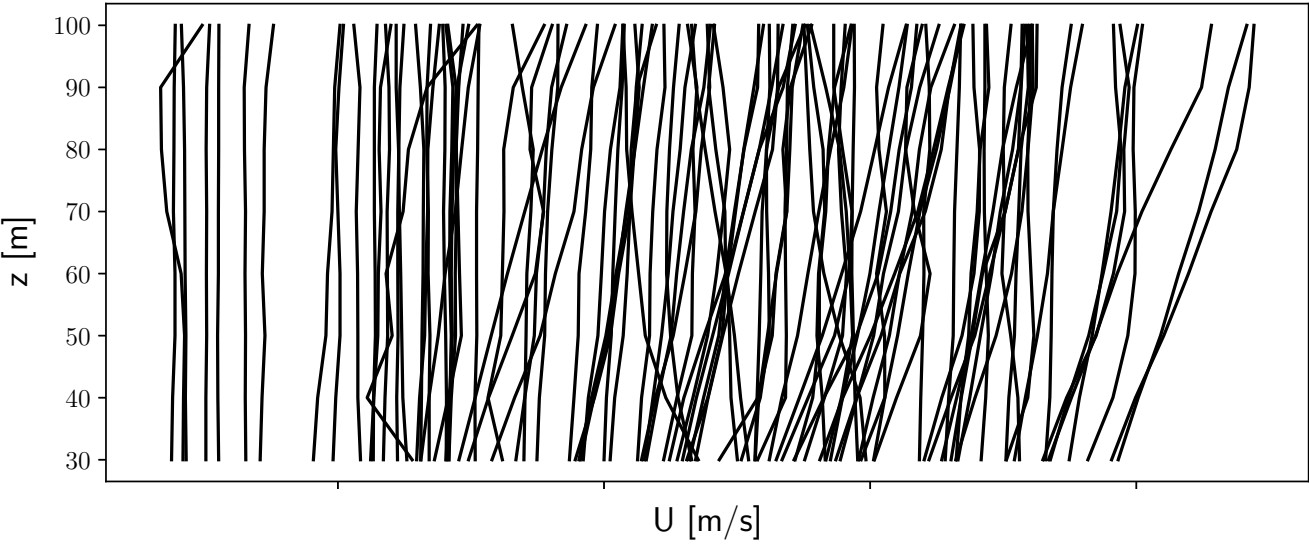

**Figure 2.** 10-minute averaged vertical wind profiles at FINO2, where the wind speed at the lowest measuring point is shifted by 0.3 m/s per profile. Each profile is taken 20 hours after the previous.

which had 11 measurement heights. The exact measurement heights at each site used is given in Table 1. The sites were chosen based on data availability and location, with the aim of having data sets of high quality and a diversity of locations. Of the seven sites, 3 were located offshore (FINO1, FINO2 and FINO3) in the North or Baltic Sea with varying distance to shore, and 4 onshore (Skipheia, Høvsøre, Valsneset and Ryningsnäs). The location of each site can be seen in Fig 3 and Fig. 4. Of the

5 4 onshore sites, Skipheia and Valsneset are located in direct proximity to the Norwegian Sea and were therefore additionally categorized as coastal. Høvsøre is in this study occasionally referred to as semi-coastal since it is located only 1.7km from the Danish North Sea coastline and mainly experiences offshore incoming winds. Ryningsnäs is located in a forested region in Sweden 30km inland.

The time periods of the data recordings as well as the data availability after filtering the data according to the method

described in Section 3.2 can be seen in Table 2. The data coverage was for all sites except Valsneset at least one year, and for many sites covered several years, yielding a robust framework for conducting a thorough analysis. At the FINO sites the time periods could be chosen since the measurements were downloaded from an online interface. The periods were all chosen to be early in the mast lifetime due to high data availability and simultaneously avoiding distortion from the construction of nearby wind farms. The measurement data from all sites was provided and analyzed in the form of time-stamped 10-minute averages.

At the FINO sites and at Ryningsnäs, the wind speed heights were for visualization purposes named and visualized according to their nearest number divisible by 10, since some measurements had slight offsets (i.e 32 is mentioned as 30, 51 as 50. See Table 1 for exact measurements). Section 3.1 provides a more detailed description of each site.



| Site | Measurement | Height [m] | Removed |
|---|---|---|---|
| **Skipheia** | Wind speed | 10, 16, 25, 40, 70, 100 | - |
| | Wind direction | 10, 16, 25, 40, 70, 100 | - |
| | Temperature | 0.2, 10, 16, 25, 40, 70, 100 | - |
| | Relative humidity | *Extrapolated from nearby source* | - |
| | Pressure | *Extrapolated from nearby source* | - |
| **FINO1** | Wind speed | 33, 42, 52, 62, 72, 82, 92, 103 | - |
| | Wind direction | 33, 51, 71, 91 | 51, 71 |
| | Temperature | 33, 42, 52, 72, 101 | |
| | Relative humidity | 34, 42, 52, 72, 101 | 42, 72 |
| | Pressure | 21, 92 | 92 |
| **FINO2** | Wind speed | 32, 42, 52, 62, 72, 82, 92, 102 | - |
| | Wind direction | 31, 51, 71, 91 | - |
| | Temperature | 30, 40, 50, 70, 99 | - |
| | Relative humidity | 30, 50, 99 | - |
| | Pressure | 30, 90 | - |
| **FINO3** | Wind speed | 31, 41, 51, 61, 71, 81, 91, 101 | - |
| | Wind direction | 29, 101 | - |
| | Temperature | 29, 55, 95 | 95 |
| | Relative humidity | 29, 55, 95 | - |
| | Pressure | 23, 95 | - |
| **Høvsøre** | Wind speed | 10, 40, 60, 80, 100, 116.5 | - |
| | Wind direction | 10, 60, 100 | - |
| | Temperature | 0, 2, 100 | - |
| | Relative humidity | 2, 100 | - |
| | Pressure | 2, 100 | - |
| **Valsneset** | Wind speed | 40, 50, 60, 70, 80, 90, 100, 110, 120, 130, 140 | - |
| | Wind direction | 40, 50, 60, 70, 80, 90, 100, 110, 120, 130, 140 | - |
| **Ryningsnäs** | Wind speed | 40, 59, 80, 98, 120, 137.7 | - |
| | Wind direction | 40, 59, 80, 98, 120, 137.7 | - |
| | Temperature | 40, 59, 80, 98, 120, 137.7 | - |
| | Pressure | - | - |
| | Relative humidity | - | - |

**Table 1.** Site instrumentation with measurement heights at each site. The removed quantities are explained for the individual sites in Section 3.1.

**Figure 3.** Map showing location of all sites used in this study.

## 3.1 Measurement sites

### 3.1.1 Skipheia (Frøya)

The Skipheia meteorological mast is operated by NTNU, and located at the western mid-Norway coast on the island of Frøya. The mast is located on land, approximately 20m above sea level and with the shortest distance to the ocean being 300m in the south/southwestern direction. The site experiences winds coming in from the Norwegian sea from the south-west, as well as onshore winds from the east. The site has 6 measurement heights from 10-100m of wind velocity, direction and temperature,



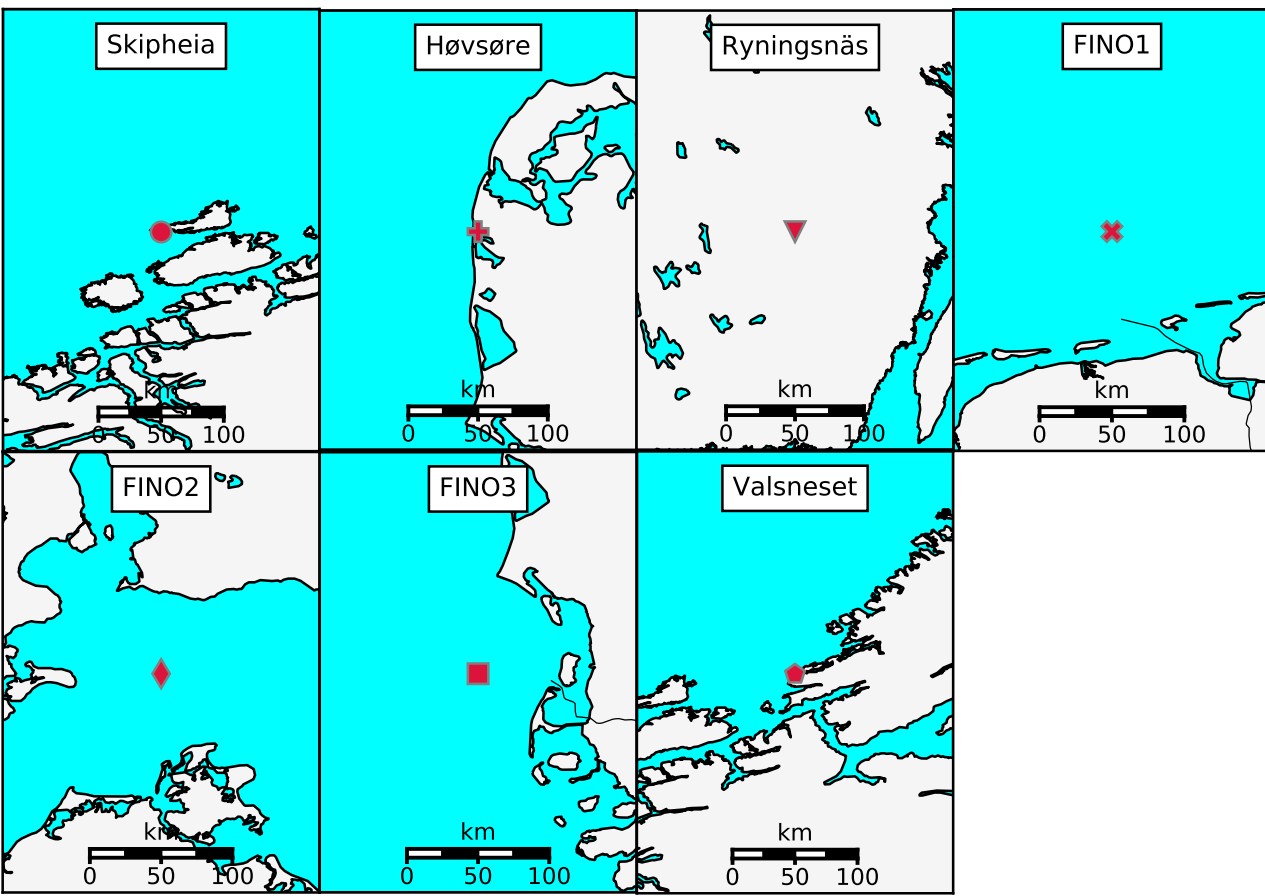

**Figure 4.** Close up of each site location.

and an additional near-ground temperature measurement. The data is available online, where more information can be found in Domagalski and Sætran (2019). The wind velocity is recorded by two 2D ultrasonic anemometers at each height mounted in two oppositely facing directions, the wind measurement not in the mast shadow was used at each time interval. The site does not record pressure or humidity, this was extracted from a nearby meteorological station for usage in the stability analysis. The uncertainty of this is discussed in 3.3. The Skipheia site had notable downtime during the measurement campaign but the length of the campaign ensured data coverage across all times of year which had a recording time equivalent of well over one year of measurements.

### 3.1.2 FINO1

The FINO1 site is located in the German Bight (North Sea) approximately 45km north of the island of Borkum. The distance to shore varies largely with direction as seen in Fig. 3. The FINO1 data was made available through personal communication





| Site | Time-period | Distorted sector | Removed data |
|---|---|---|---|
| Skipheia | 14.12.2009 - 22.11.2014 | - | 69.22% |
| FINO1 | 01.01.2005 - 31.12.2006 | - | 30.01% |
| FINO2 | 01.01.2010 - 31.12.2012 | - | 11.02% |
| FINO3 | 01.01.2010 - 31.12.2012 | - | 26.96% |
| Høvsøre | 01.01.2017 - 31.12.2017 | $290° - 45°$ | 31.36% |
| Ryningsnäs | 02.11.2010 - 04.02.2012 | $30° - 100°, 120° - 220°, 305° - 360°$ | 60.11% |
| Valsneset | 06.01.2014 - 22.09.2014 | $356° - 144°$ | 67.56% |

**Table 2.** Time period of data extraction, the distorted sectors at each site, and the percentage of data removed after filtering according to section 3.2.

with personnel at The Federal Maritime and Hydrographic Agency (BSH), and when data access was granted data from the entire measurement campaign at FINO1 was available to download through an online interface. Further information regarding the FINO1 site and instrumentation can be found in FINO1 (2019).

The FINO1 site has a research mast that is highly equipped with both temperature, wind speed and wind direction measurements. The cup anemometer measurements used in this study measure the wind speed at 8 heights from 30-100m, which are mounted on one boom of length 3.0-6.5m at each height, and the booms are mounted in the directional sector $135° - 143°$. The top anemometer is mounted on top of the mast in a lightning protection cage. The relative humidity measurements at z=42m and 72m, as well as the pressure measurement at z=92 had large data gaps and were not used for the entire study (see Table 1).

### 3.1.3   FINO2

The FINO2 data was made available through an online database in the same fashion as the FINO1 data, see Section 3.1.2. The FINO2 site is offshore, located in the southwestern part of the Baltic Sea approximately 33km north of the German island of Rügen. The site experiences a mixture of fetch distances, being located within the triangle of Denmark in the west, Sweden in the North and Germany in the south. Cup anemometers measure the wind speed at 8 heights from 30-100m, and from one direction at each height ($180°$). The top anemometer is mounted on top of the mast in a different fashion to the other wind speed
measurements. The data set had few gaps an a high availability which can be seen in Table 2. Further information regarding the FINO2 site can be found in FINO2 (2019).

### 3.1.4   FINO3

The FINO3 data was made available through an online database in the same fashion as the FINO1 data, see Section 3.1.2. The FINO3 site is located north of FINO1 in the German Bight (North Sea), approximately 80km west from the German
island of Sylt. The site is instrumented with several booms to account for flow distortion, however not at all heights. Wind speeds recorded on booms in the direction $345°$ were used for all 8 measurements heights from 30-100m for consistency. At





FINO3 the temperature at z=95m was found to be missing when downloading the data regardless of the period chosen, and was therefore not used in the final analysis. Further information regarding the FINO3 site can be found in FINO3 (2019).

### 3.1.5 Høvsøre

One year of data from the Høvsøre meteorological mast was made available through personal communication with Yoram Eisenberg of DTU Wind Energy. The Høvsøre site is located at the west coast of Denmark in the coastal farmland of west Jutland. The site is located in a flat area and homogeneous area, the surrounding features include the village of Bøvlingbjerg approximately 3km southeast, the North Sea coastline with a sand embankment 12m high 1.7km west, and the Nissum Fjord 800m to the south. The site conducts tests on several masts and turbines, and the measuring mast used in this study is located directly south of a row of 5 turbines which are aligned in northern direction, and each of these turbines is additionally paired with a power mast located 200m west of it (Smith et al., 2006). The measurements used in this study are recorded by a meteorological mast where the wind velocity is recorded at 6 heights from 10-116.5m (see Table 2). The cup-anemometers and wind vanes are all installed on south facing booms, thus making the mast distortion in the same direction as the turbine wake influenced region. A 115° sector was excluded to avoid turbine distortion, as well as distortion due to the power masts. This was a conservative approach in comparison to the recommended practice (IEC, 2017) and the common practice when analyzing the Høvsøre data which is to disregard mast effects (Peña et al., 2016). Further information on the site as well as results on 10 years of measurements at the Høvsøre site has been published by Peña et al. (2016).

### 3.1.6 Ryningsnäs

The Ryningsnäs data was made available through personal communication with Johan Arnnqvist at Uppsala University. Ryningsnäs is a forested location in Sweden, approximately 30km inland from the Swedish southeastern coast (Arnqvist et al., 2015). The terrain in the region is mostly flat with mild variations, due to forestry and natural variations the landscape is however not completely homogeneous. The measurements are conducted through equipment installed on a 140m high mast located in the northwestern corner of a 200x250m clearing. The wind velocity was recorded at 6 heights on the mast by 3D ultrasonic anemometers. Two turbines are present at the site approximately 200m from the mast in the southern and northeastern direction respectively. The sectors affected by the nearby turbines as well as the mast (mast effects were observed) were removed in the analysis, the sectors are given in Table 2. The pressure and relative humidity were not measured at the mast and an analysis of atmospheric stability at Ryningsnäs was therefore not conducted. Further information on the Ryningsnäs site is given by Arnqvist et al. (2015).

### 3.1.7 Valsneset

The Valsneset site is located northwest of Trondheim (Norway) on the peninsula of Fosen. The data was made available through personal communication with Lars Morten Bardal at NTNU. The site is situated in immediate vicinity to the Norwegian Sea in the north and west, and with a mixture of smaller and bigger rocks as well as sea in the south and east. The data used originates



| | No maximum | 1 maximum | 2 maxima | 3 maxima | 1 minimum | Reversed |
|---|---|---|---|---|---|---|
| **Skipheia** | | | | | | |
| Number of cases | 49 855 | 26 695 | 3 161 | | 2 402 | 750 |
| Percentage | 60.17% | 32.22% | 3.81% | | 2.90% | 0.90% |
| **Høvsøre** | | | | | | |
| Number of cases | 30 195 | 5 139 | 367 | | 147 | 53 |
| Percentage | 84.11% | 14.31% | 1.02% | | 0.41% | 0.15% |
| **Ryningsnäs** | | | | | | |
| Number of cases | 22 574 | 2 649 | 267 | | 115 | 10 |
| Percentage | 88.13% | 10.34% | 1.04% | | 0.45% | 0.04% |
| **FINO1** | | | | | | |
| Number of cases | 16 732 | 34 884 | 19 041 | 737 | 1 823 | 361 |
| Percentage | 22.74% | 47.41% | 25.88% | 1.00% | 2.48% | 0.49% |
| **FINO2** | | | | | | |
| Number of cases | 47 236 | 58 267 | 24 631 | 3 522 | 5 514 | 1 138 |
| Percentage | 33.67% | 41.53% | 17.55% | 2.51% | 3.93% | 0.81% |
| **FINO3** | | | | | | |
| Number of cases | 38 718 | 47 685 | 25 220 | 1 872 | 1 283 | 490 |
| Percentage | 33.59% | 41.37% | 21.88% | 1.62% | 1.11% | 0.43% |
| **Valsneset** | | | | | | |
| Number of cases | 7 050 | 3 561 | 758 | 83 | 404 | 239 |
| Percentage | 58.27% | 29.43% | 6.26% | 0.69% | 3.34% | 1.98% |

**Table 3.** Occurrence of different profile categories at all sites. Blank spaces indicate that the site had too few measurement heights for the profile category to be possible.

from a Lidar measurement campaign which ran for 10 months, and measured wind speeds at 11 heights of 10m-increments from 40-140m. The data availability was set to a requirement of >99% in each 10-minute recording interval to ensure correct 10-minute averages. The lack of temperature measurements prohibited a stability analysis at Valsneset. The site has several nearby wind turbines restricting the wind sector analyzed (see Table 2) which was removed following the recommendation of
5 IEC (2017). A more detailed description of the Valsneset site is given by Bardal et al. (2015).

## 3.2 Data filtering

For all sites, the time series of the 10-minute averaged data was filtered to remove any non-physical measurements as well as wind data from distorted directional sectors if the site exhibited wind distortion. Non-physical measurements entailed measurements which were artificially high or low compared to their typical range. In addition the FINO1, FINO2, FINO3 and
10 Ryningsnäs data sets were obtained with attached quality tags at each time-level, any data entry tagged as poor was therefore





removed. The distorted sectors of a site are described in their respective site section and an overview is provided in Table 2. The effect of mast distortion is discussed in Section 3.2.1.

After the tagging of poor measurements, the data removal was done as follows: if any measurement (direction, velocity, temperature, pressure, relative humidity) was missing due to downtime, from a distorted direction, found to be non-physical
or tagged with a poor quality, all data from this 10-minute interval was removed and all measurements within this 10-minute average was therefore discarded. Some measurements at FINO1 and FINO3 did however have longer periods of downtime which impacted the filtering to such a degree that they had to be removed, an issue which was similarly encountered by Argyle and Watson (2014) at FINO3. The removal of a quantity was only done if the measurement was not a wind speed measurement, and if the same quantity was available at other heights so that its removal did not restrict any additional analysis. The removed
quantities are given in Table 1. No filter was set with regards to minimum velocity of the data, or to the standard deviation of a 10-minute averaged quantity. This was done intentionally to avoid the results being artificially affected by these filters, which was also done in the similar study by Kettle (2014).

### 3.2.1    Filtering mast distortion

Wind measurements from meteorological mast may from certain sectors be affected by the the mast itself, a phenomenon called
mast distortion. A common way of circumventing this is to record the wind speed on booms positioned in opposite directional sectors such that there is always a direction of measurement not impacted by mast distortion. This method was enforced at Skipheia, which had wind velocity measurements in two opposite directions at all measurement heights. At Høvsøre the mast-distorted sector coalesced with a turbine-distorted sector and was subsequently removed. The Ryningsnäs data was recorded by ultrasonic anemometers which were found to show mast distortion effects, the mast-distorted sector was therefore removed.
As the Valsneset data was measured by a Lidar device a mast-distortion analysis was not necessary.

The data from the FINO sites is managed by the DEWI group which provide mast corrected wind speeds based on a uniform ambient flow correction (UAM) algorithm (Westerhellweg et al., 2012). Mast corrected wind speeds were however only available at all heights at the FINO2 site. The analysis in this study was conducted on both the mast-corrected and non-corrected wind speeds at FINO2, as well as both including and excluding the mast-distorted sectors at all FINO sites. The
results were found to be similar in all cases, except for a large number of inflections at the uppermost height when using the mast-corrected wind speeds at FINO2. The mast-corrected winds speeds are however computed with some uncertainty, and since the mast-corrected wind speeds were only available at all heights at FINO2, the regular uncorrected wind speeds were used in this study. Since mast effects were found to be negligible when excluding the distorted sector, no mast distortion filtering was employed. The same conclusion was drawn by Kettle (2014) when studying local maxima at FINO1.

### 30    3.3    Atmospheric stability calculation

A part of this study includes the investigation of the correlation between atmospheric stability and abnormal vertical wind profiles. The stability analysis was conducted using the Richardson number (Arya, 1988) to calculate the Obukhov length and subsequently dividing the occurrences into the 5 stability classes (very stable, stable, neutral, unstable, very unstable) using





Obukhov length bins given by Bardal et al. (2018). Certain sites in the analysis had ultrasonic anemometers which would have enabled a sonic method of stability calculation. There were however issues with data gaps in the ultrasonic measurements at the FINO sites, as well as ultrasonic anemometers not being installed at all sites. The Richardson method was therefore employed, which excluded Ryningsnäs and Valsneset from the stability analysis due to lack of measurements. The remaining

sites were however found to sufficiently describe effect of atmospheric stability on abnormal vertical wind profiles. At sites where the pressure or relative humidity were only available at one height they were assumed constant. When relative humidity was available at two heights but not the height of the temperature measurement it was linearly interpolated. The effect of varying the relative humidity was tested and did not change the conclusions of the study but may be a source of uncertainty in the stability analysis, especially during neutral conditions (Peña et al., 2008). The effect of only having a pressure measurement

at one height was tested and found to be minimal, the same conclusion was drawn by Argyle and Watson (2014).

For the offshore sites the gradient Richardson formulation was used due to low availability of sea temperature measurements, while for the onshore sites near-ground measurements enabled use of the bulk Richardson number formulation. While the gradient method provides a more correct description of the dynamics of the boundary layer, it requires careful calibration of the instrumentation. The gradient method is due to the postulated thermal layering of the MABL also found by Argyle

and Watson (2014) to be dependant on the measuring heights used. There is therefore a degree of uncertainty related to the stability analysis, at several sites the stability distribution was therefore compared to previous studies and showed reasonable agreement (Høvsøre: Peña et al. (2016), Skipheia: Bardal et al. (2018), FINO1 and FINO3: Argyle and Watson (2014)). The use of different measurement heights was in addition thoroughly tested at FINO1, FINO2 and Skipheia, and although the stability distributions showed variation, the same tendencies prevailed and the same conclusions were drawn regardless of the

measurement heights used.

## 4 Results

To illustrate the variation in the 10-minute averaged vertical wind profile, a selection of arbitrary profiles from FINO2 are plotted in Fig. 2. The profiles clearly illustrate that the 10-minute averaged wind profile does not necessarily conform with the shape of neither the power law nor the logarithmic law. Some profiles represent instances where the wind increases with height

as expected, a significant amount of the profiles do however exhibit unexpected traits of singular or multiple local maxima. Figure 2 clearly demonstrates the importance of resolving issues associated with the vertical wind profile description.

The quantity in occurrence of the different profiles categorized by the number of local maxima is in this study the primary indicator of the in-applicability of the commonly used wind profile formulations. The percentage-wise and total occurrence of the different profile categories is presented in Table 3 and as a histogram in Fig. 5. The blank spaces indicate that the site

has too few measurement heights to experience such a number of local maxima (also referred to as 'kinks' or 'inflections'). Valsneset is the only site which had enough measurement heights to record instances of 4 local maxima, the occurrence was however as low as 0.03% (4 cases) and is excluded from Table 3.

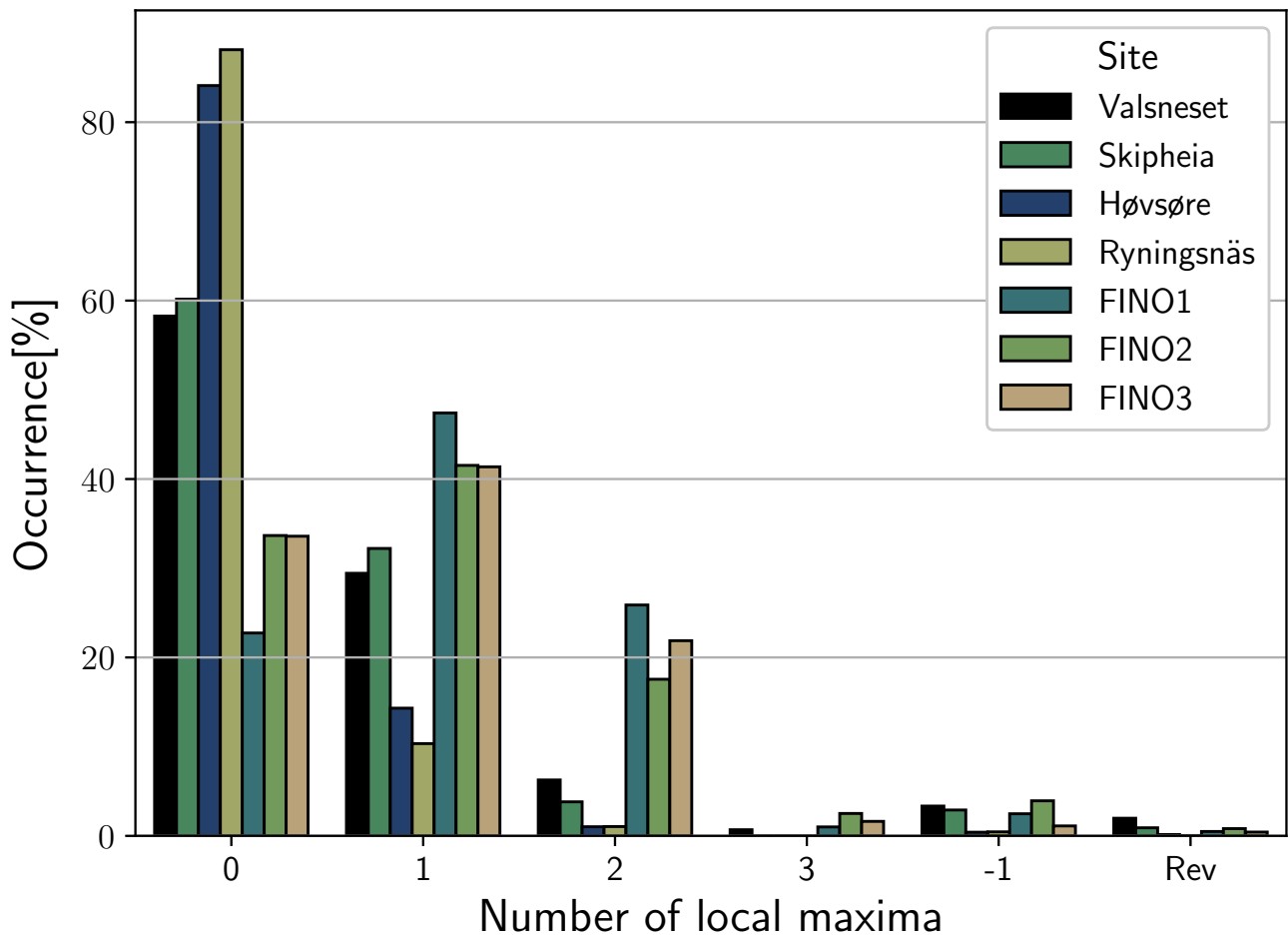

**Figure 5.** Histogram of profile categories occurrence by number of local maxima.

The results reveal that offshore sites are most prone to abnormal vertical wind profiles and therefore have the lowest occurrence of the expected 0-inflection vertical wind profile. At the offshore sites FINO1, FINO2 and FINO3 profiles are found to be predominantly abnormal and profiles exhibit inflections or a reversed profile in 77.26% of profiles at FINO1, 66.33% at FINO2 and 66.41% at FINO3. The onshore occurrence of abnormal profiles is found to scale inversely with the distance to shore, and

5 the two coastal sites Skipheia (39.83%) and Valsneset (41.73%) therefore both show a higher occurrence of abnormal profiles amongst the onshore sites. This decreases for the semi-coastal site Høvsøre (15.89%) and abnormalities are most rare for the far-inland site Ryningsnäs (11.87%). Although the three FINO sites have different fetch distances ranging from 30-80km and being located in different offshore conditions (North/Baltic Sea) this is not displayed in the results, and analysis of several years of data from the sites did not reveal a correlation between the fetch of a site and the occurrence of local maximum in the



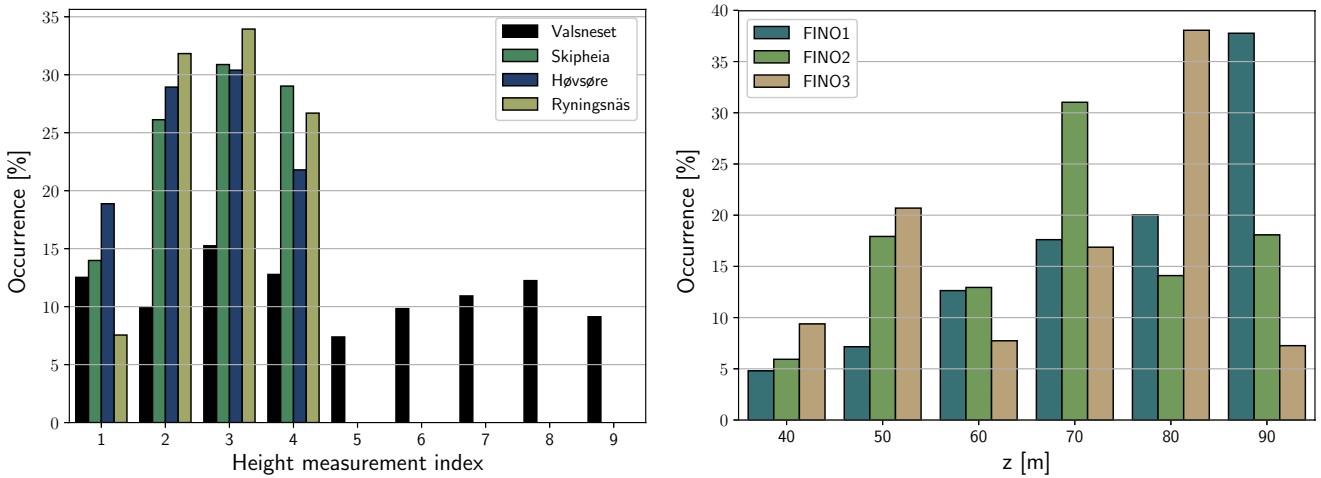

**Figure 6.** Histogram showing the height at which the inflection occurs for the 1-inflection profiles at onshore and offshore sites. In the left figure the x-axis is the height index, with 1 being the second measurement height. Table 1 provides the measuring heights used at all sites.

vertical wind profile. As the North Sea and Baltic Sea, where the FINO sites are located, are both to a varying degree enclosed by land it is unclear whether this result holds for winds which are clear of coastal effects.

When abnormalities are present they are predominantly in the form of 1 local maximum. The 1-inflection profiles are found to occur most often at FINO1 (47.41%), with FINO2 (41.53%) and FINO3 (41.37%) showing slightly lower and similar relative
occurrences. The 1-inflection profiles are also the dominant abnormal profile type onshore, being present in 29.43% of profiles at Valsneset, 32.22% of profiles at Skipheia, 14.31% of profiles at Høvsøre and the lowest occurrence of only 10.34% of profiles at Ryningsnäs. Here a scaling was found, namely that, of the onshore sites, the coastal sites have a higher amount of profiles with multiple inflections, while further inland the 1-inflection category becomes more common among the abnormal profiles.

The results also reveal that the three offshore FINO-sites exhibit the most amount of profiles with 2 local maxima. All FINO
sites have a percentage-wise higher occurrence of 2 local maxima (FINO1: 25.88%, FINO2:17.55%, FINO3: 21.88%) than the coastal Valsneset site (6.26%), even though the amount of measurement heights is 11 at Valsneset and 8 at the FINO sites which makes a profile with several local maxima more probable at Valsneset. This clearly indicates that local maxima are more prominent at offshore sites than they are onshore. The 3-inflection profile occurs rarely and therefore does not have large implications for wind energy applications. It is also mentioned here that the results of FINO1 as expected very similar to the
results found by Kettle (2014) who studied local maxima in the VWP for the year 2005 at FINO1.

The reversed and -1-inflection profiles also occur at all sites, but similar to the 3-inflection profiles their occurrence is too low to be very relevant for wind energy applications. It should however be mentioned that that the occurrence of these categories is linked; a site with a higher amount of reversed profile is also seen to have a higher occurrence of -1-inflection profiles.

From these results it is clear that abnormalities are most common offshore, and are found to decrease with an increasing
surface roughness. For onshore sites, locations in direct proximity to the coast (such as Skipheia and Valsneset) are found





to be much more prone to abnormalities than sites only a few kilometers inland (Høvsøre). The increased surface roughness associated with the forested site Ryningsnäs results in higher degrees of turbulence which leads to large mixing and less abnormal profiles. This is seen to correspond to a low occurrence of inflected vertical wind profiles.

### 4.1 The effect of using only 4 measurement heights

5  Evidently the probability that the vertical wind profile contains one or more local maxima increases with the number of measurement heights. In addition the varying height increment between measurements can cause differences in the occurrence of local maxima. The vertical wind profiles were therefore analyzed using only the heights z=(40m, 60m, 80m, 100m), which are approximately common for all sites except Skipheia. At Skipheia the closest replication of this was used, namely z=(25m, 40m, 70m, 100m). With 4 measurement heights the possible profile categories are: 0-inflection, 1-inflection, -1-inflection and 10  reversed.

| Inflections: | 0 | 1 | -1 | Rev. |
|---|---|---|---|---|
| **Skipheia** | 67.11% | 26.89% | 3.36% | 2.64% |
| **Høvsøre** | 87.42% | 9.24% | 1.71% | 1.63% |
| **Valsneset** | 76.00% | 15.95% | 2.99% | 5.06% |
| **Ryningsnäs** | 94.14% | 5.11% | 0.58% | 0.17% |
| **FINO1** | 46.57% | 40.34% | 10.72% | 2.36% |
| **FINO2** | 58.71% | 27.69% | 9.31% | 4.29% |
| **FINO3** | 58.75% | 25.83% | 12.54% | 2.87% |

**Table 4.** Local maxima results if all sites are restricted to only 4 measurement heights. At all sites except Skipheia, the common heights z=(40m, 60m, 80m, 100m) were used, at Skipheia, z=(25m, 40m, 70, 100m) were used.

The results in Table 4 confirm that offshore sites to a larger degree experience local maximum in the wind profile than onshore sites. Using only 4 heights results in an increase in the 0-inflection profile at all sites, but the differences between the sites remains the same and similar conclusions on the abnormal profile occurrence can be drawn. Since the use of all measurement heights does not alter the main conclusions all heights were used for the remainder of the study.

15  ### 4.2 Height of local maximum

The height at which the wind profile deviates from its expected shape is essential when assessing the impact the inflected profiles have on a wind turbine. Local maxima at wind turbine hub heights were shown by Wagner et al. (2009) to significantly impact the power output due to the impact the negative shear has on the available energy across the rotor area. The inflections can however also have positive consequences as Gutierrez et al. (2017) found the negative shear in the top half of a low-level 20  jet to dampen motion, forces and moments acting on the turbine tower and nacelle.



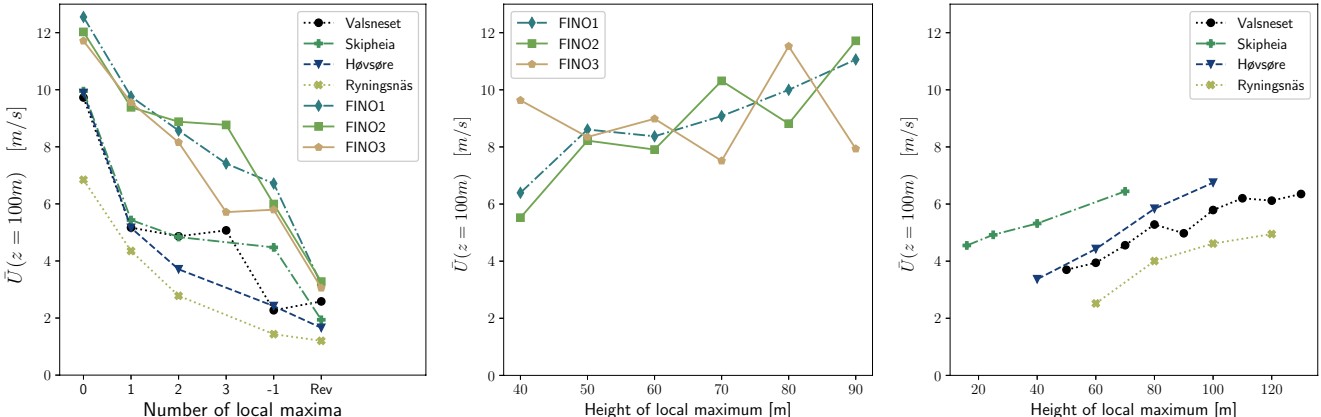

**Figure 7.** Left: Mean wind speed at z=100m of different profile categories. Middle: Mean wind speed at z=100m for offshore sites, with VWP maxima at different heights. Right: Mean wind speed at z=100m for onshore sites with VWP maxima at different heights. Similar results where found when using the lowest wind speed measurement height at each site.

### 4.2.1 Onshore sites

The left histogram in Figure 6 shows the height occurrence of local maximum at the onshore sites. Comparison of the onshore maxima heights is complicated due to the variation in the number of measurement heights and the difference in height increments, and was therefore visualized as a function of the height index of the local maxima.

At all onshore sites except Valsneset the occurrence of an inflected profile is found to increase with the height of the inflection up to the second highest measurement height. From the second highest to the highest point, all these sites show a slight decrease in occurrence. The reason for the slight decrease at the uppermost height could be related to a speed-up effect at the top measuring point of the mast, which would make top-layer inflections less common.

   At the coastal site Valsneset the variation of local maxima occurrence stands out in comparison to the other sites. At the

lower heights of z=(50m, 60m, 70m, 80m) the occurrence seems arbitrary and evenly distributed. After this the occurrence of local maxima decreases for z=90m before showing the same monotonic increase in occurrence with height followed by a top height decrease as was found at the other onshore sites. The twofold variation found at Valsneset is not entirely clear, but could be caused by a transition from the surface layer to the Ekman layer at an intermediate height, which may not be visible at the similar site Skipheia, due to lower measuring heights. The aforementioned top height decrease is also visible at Valsneset but

is not due to mast speed up since the measurements were performed by a Lidar device. The cause of this is result is not entirely clear but at the masted sites speed-up effects can not be excluded as the cause.





### 4.2.2 Offshore sites

The right histogram in Figure 6 shows the height of the local maximum at offshore sites for the 1-inflection case. The results show that inflections occur at all heights, but the percentage of occurrence at each height varies to a larger extent than onshore. At FINO1 this variation is however not present, and the occurrence is seen to consistently increase when the inflection occurs

higher in the profiles. At FINO2 the profiles are most commonly inflected at the third highest inflection height z=70m, and a slight general increase in occurrence with height is visible. The profiles at FINO3 show the largest variation, and are found to be most commonly inflected at the second uppermost height z=80m but least commonly inflected at the top height z=90m.

The variation in inflection height occurrence at the offshore sites is partially explained through smaller height increments between measurements ($\Delta z \approx 10m$). The occurrence at each height is however also found to be strongly coupled with the

atmospheric stability, as the heights with the highest occurrence of inflections are the heights with the highest amount of very unstable atmospheric conditions. This is discussed in Section 4.4.2.

In summary there does seem to be indications of local maxima occurring more frequently at higher elevations both onshore and offshore, with this result being more clearly visible at the onshore sites. The increased occurrence of local maxima at higher elevations should be a concern as these elevations are within common rotor swept areas of modern turbines and can

have a direct impact on the available energy in the wind. The impact on the turbine loads may however not be only negative, as Gutierrez et al. (2017) showed that a negative shear dampened motion, moments and forces acting on the turbine tower and nacelle.

### 4.3 Correlation to wind speed

It is of interest to describe the atmospheric conditions which cause abnormal vertical wind profile development. The wind

speed is a vital part of the atmospheric conditions, as well as being the source of energy in wind turbine power generation.

### 4.3.1 Wind speed relationship to profile categories

The left plot in Figure 7 shows that the mean wind speed is decreasing with an increasing number of local maxima in the vertical wind profile and at its lowest during instances of a reversed profile or a -1-inflection profile. These profiles are rare instances which only comprise a few percent of the profile cases, and their low wind speeds make them even less relevant

for wind energy applications. In the remaining profile categories, the mean wind speed is for the offshore sites well above a typical cut-in speed of 4-5 m/s, making them relevant in wind energy extraction (Cooney et al., 2017). At the coastal sites Skipheia and Valsneset the mean wind speed of the abnormal profiles lies in the approximate region of typical cut-in wind speeds, at the semi-coastal site Høvsøre and the inland forested site Ryningsnäs the mean wind speed in the abnormal profiles is however seen to decrease to below typical cut-in speeds. To investigate the impact of these abnormal profiles, the spectrum

of velocity within which the profiles occur was checked. The results showed that the range of wind speeds associated with a profile category only changes slightly from the 0-inflection to the 1-inflection profiles, and at the offshore sites only slightly from the 1-inflection to the 2-inflection category. This entails that the decrease in mean wind speed is predominantly due to





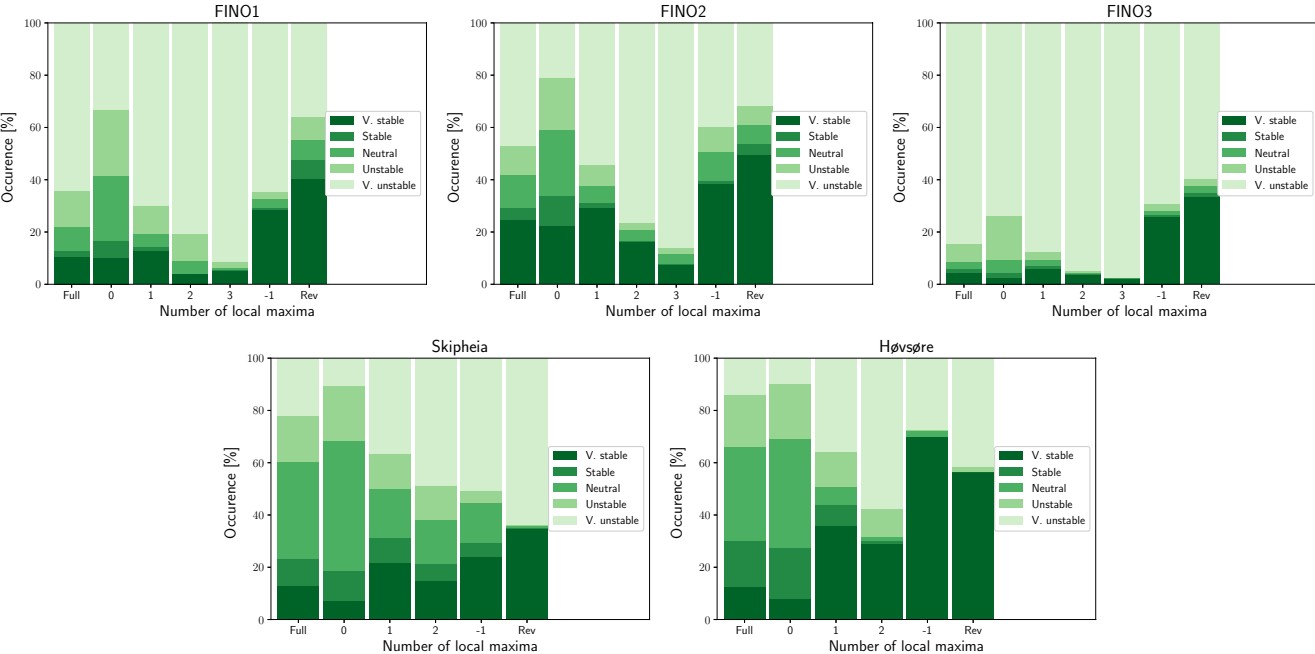

**Figure 8.** Stability distribution of profile categories at the 5 sites which had measurements enabling this analysis. Full indicates the entire data set with no abnormal profile categorization.

the peak in distribution being shifted to lower velocities. Therefore, although many of the abnormalities at the onshore site occur below cut-in wind speeds, there are still instances where these inflections are relevant for wind energy extraction. At the offshore and coastal sites the mean wind speed is higher and most of the inflected profiles will have an impact on the available energy. It is also worthwhile to consider that when Lange et al. (2004) studied the effect of erroneous offshore vertical wind

extrapolation methods on the error in predicted power output at a hub height of z=50m the errors were largest at wind speeds between 5-9 m/s. Therefore, even if an inflected profile has a wind speed only slightly above cut-in wind speed it is still relevant for wind engineering purposes.

### 4.3.2   Wind speed relationship to maxima at different heights

The wind speed at z=100m as a function of inflection height is also shown in Fig. 7. At the onshore sites (right plot, Fig. 7) a

maximum at a higher altitude is seen to correspond with higher wind speeds, with all sites exhibiting a somewhat consistent increase in mean velocity with inflection height. At the offshore sites (middle plot, Fig 7) the results shows an increase in wind speed with height, there is however larger variation between the heights. The variation is especially prominent at FINO3, and slightly prominent at FINO2. The variation was found to be partially but not entirely caused by the smaller height increment of $\Delta z \approx 10m$ at the FINO sites. The complete picture of why this is occurring is strongly coupled to atmospheric stability,

and specifically the the higher degree of very unstable inflections found at the offshore FINO sites. The inflections during very




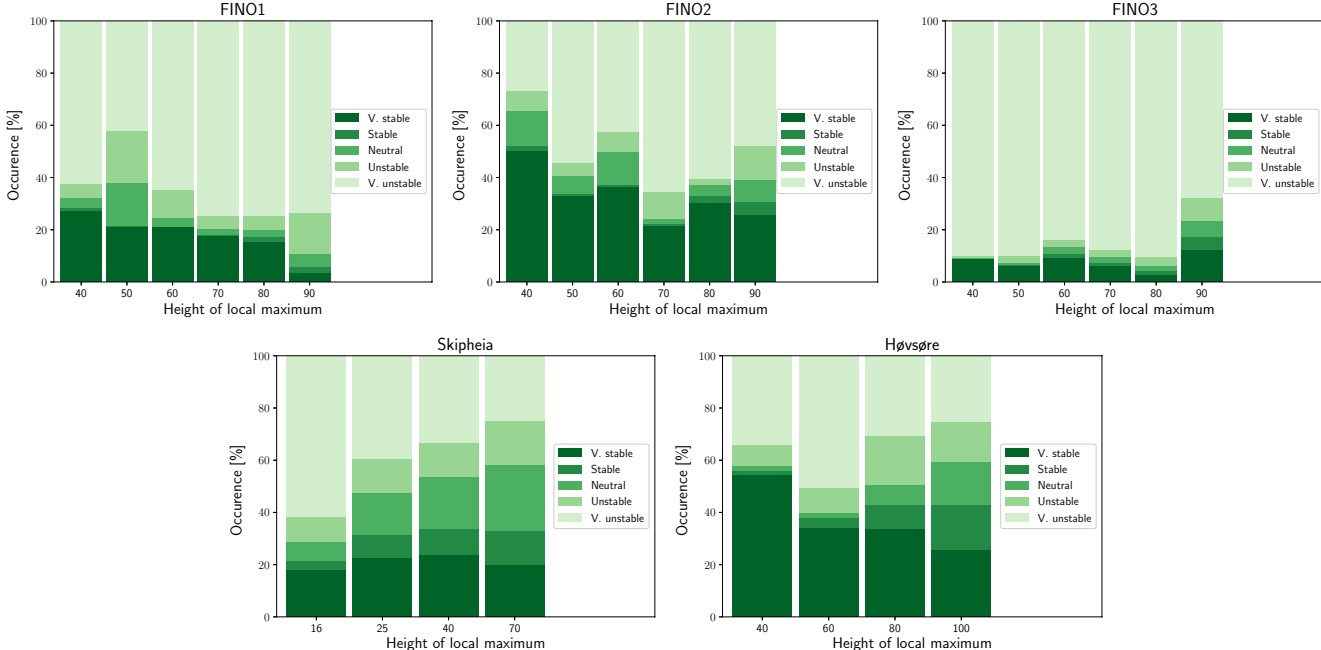

**Figure 9.** Stability distribution of 1-inflection profiles with varying height of inflection at the 5 sites which had measurements enabling this analysis.

unstable conditions show significantly larger variation in wind speed with a changing maximum height, which can be seen at FINO3 (the site with the largest variation) by the order from left to right of the profiles in the top right plot of Fig. 10 versus the bottom right plot of Fig 10.

The general increase in velocity with maximum height may not have been expected if the inflections are assumed to be
coupled with a boundary layer discontinuity which scales inversely with velocity. The surface layer depth is however mainly decided by the thermal sate of the atmosphere, i.e the atmospheric stability, which is inherently coupled to the wind speed (Stull, 2017). As wind speeds increase the atmosphere is known to transition towards a neutral atmosphere where the surface layer height increases in comparison to stable conditions. The importance of this result therefore lies mainly in communicating that higher altitude inflections could be a large concern for wind energy purposes since they occur at higher wind speeds and
may therefore be coupled with stronger load and energy variations.

### 4.4 Correlation to stability

Atmospheric stability describes the vertical forces exerted on the parcels in the atmosphere. Put simply, during stable conditions the surface is generally cooler than the air and the buoyant forces prevent vertical motion. During unstable conditions the ground is generally warmer than the air, parcels rise and stronger turbulent mixing is observed. Neutral conditions entail a
15 thermal equilibrium where parcels are in buoyant equilibrium.





In this study the stability analysis was performed using a Richardson number formulation, and was not carried out at the sites Valsneset and Ryningsnäs due to lack of measurements. The forthcoming sections are therefore focused on the remaining sites, where FINO1, FINO2 and FINO3 are located offshore, Skipheia is coastal and Høvsøre is onshore/semi-coastal. The employed method used in the stability analysis is discussed in Section 3.3.

### 4.4.1 Stability distribution of abnormal profile categories

During the presence of local maximum in the VWP all sites shows an increasing occurrence of very unstable atmospheric conditions categorized by more vigorous turbulent mixing (Fig 8). This increase grows with the number of inflections, meaning that 2-inflection profiles have a higher occurrence of very unstable conditions than the 1-inflection profiles. This is likely linked to the flat profiles caused by this mixing state, since the inflections need less 'disturbance' or severity during very unstable conditions in order to cause a maximum in an already flat development. The two onshore sites Høvsøre and Skipheia in addition show an increase in very stable conditions where turbulence is suppressed, during the presence of one or more local maxima. These increases are seen to lead to a decrease in neutral conditions, which aligns with the decreasing wind speed for inflected profiles seen in the left plot of Fig. 7 since neutral conditions are more common at higher wind speeds. The 0-inflection category unsurprisingly shows an opposite change in atmospheric conditions to that of the inflected profiles.

The reversed and -1-inflection profile categories can at all sites be seen to occur more commonly during very stable conditions. This was found to be due to the higher occurrence of very stable conditions at the low wind speeds under which these profile categories occur.

### 4.4.2 Stability distribution with increasing maximum height

The atmospheric stability distribution for the 1-inflection profiles with a varying maximum height is shown in Fig. 9. Considering the offshore sites FINO1 and FINO2 first, the occurrence of a very stable atmosphere is higher when the maximum occurs at a lower altitude ($\approx 30\%$ and $\approx 50\%$ respectively), but decreases and is superseded by very unstable conditions as the height of the maximum increases. At FINO1 this decrease is present at all heights, and at z=90m very stable inflections are almost non-existent. At FINO2 the occurrence of very stable conditions increases again for z=80m and z=90m, and there is also a slight increase in very stable conditions at FINO3 when inflections occur at z=90m. No apparent reason for this increase was found, the possibility of these inflection heights being coupled with longer fetch distances was checked but did not yield an explanation, neither did a cross-correlation with wind speed.

Shallow surface layers and internal boundary layers are usually coupled with very stable atmospheric conditions due to the negative buoyancy forces, the higher occurrence of very stable conditions during low-altitude inflections could therefore indicate that these inflections are offshore caused by a stable internal boundary layer formation, which is often coupled with a capping inversion where larger than expected wind speed gradients may be found (Lange et al., 2004). The results of Argyle and Watson (2014) suggested an IBL formation with a height of around z=50m at FINO1 and FINO3, the higher occurrence of inflections during very stable conditions at lower altitudes supports the possibility of such an inversion.





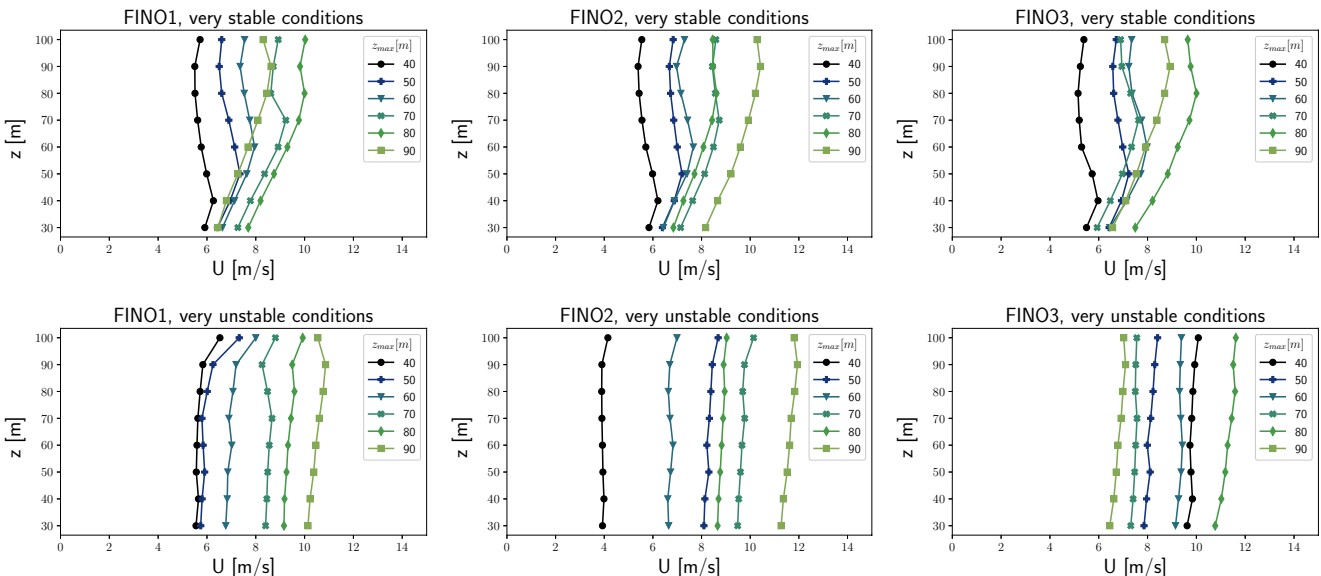

**Figure 10.** Mean 1-inflection velocity profiles at FINO1, FINO2 and FINO3 with maximum at different heights, during very stable and very unstable atmospheric stability. The neutral stability condition is not show due to a low occurrence.

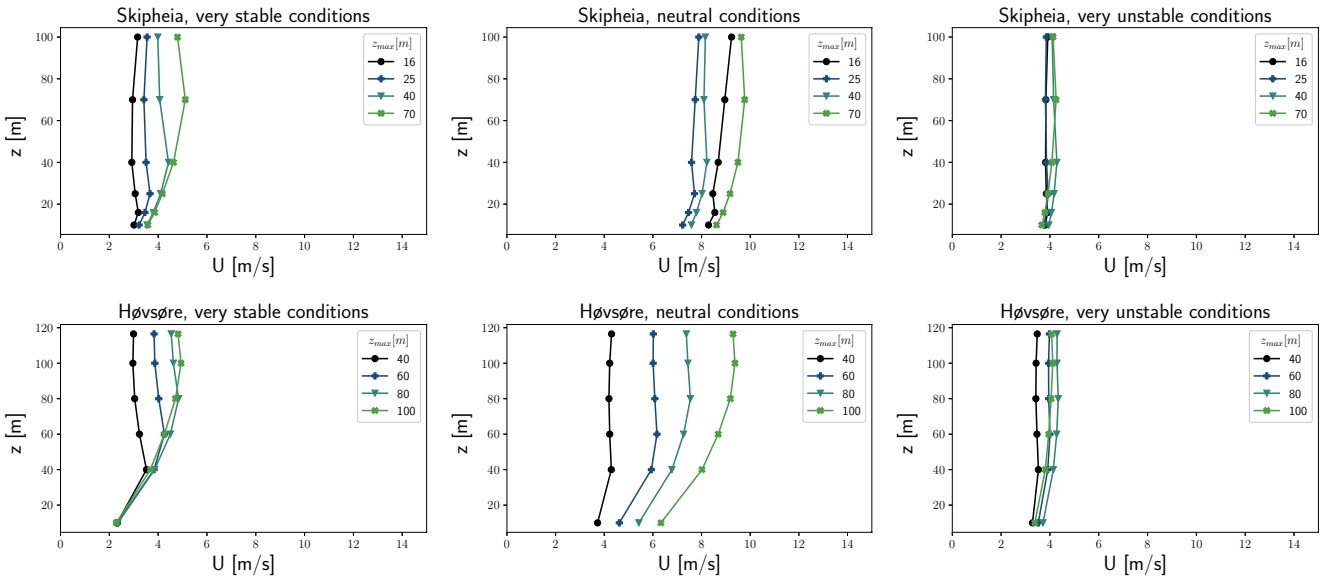

**Figure 11.** Mean velocity profiles at Skipheia and Høvsøre with maximum at different heights, during very stable, neutral and very unstable atmospheric stability.



The results at FINO3 differ from the two other offshore sites. At FINO3 the atmosphere is predominantly very unstable regardless of inflection height (top right plot, Fig. 9). Although the stability distribution at FINO3 varies minimally, the number of inflections at each height responds strongly to these small variations. The inflection height with the highest occurrence of very unstable conditions is z=80m, this is also the height with the highest amount of inflected profiles. At the opposite end of

the scale z=90m has the lowest occurrence of inflections and the lowest occurrence of very unstable conditions. Although the predominance of very unstable conditions of FINO3 was also found by Argyle and Watson (2014), those results also indicated a higher occurrence of a stable atmosphere when using temperature measurements at higher elevations. The upper temperature measurement at FINO3 (z=95m) was however unavailable in this study, the high sensitivity of the height occurrence to the atmospheric stability could therefore indicate that the FINO3 stability distribution depicts too large percentages of very unstable

conditions, and too low percentages of very stable conditions.

Onshore (Skipheia and Høvsøre) there is a decrease in very unstable conditions as the inflection rises, contrary to the offshore results. Instead there is an increase in neutral, slightly stable and slightly unstable conditions. The reason for this is related to the atmospheric conditions which prevail during the higher wind speeds. Onshore the atmosphere is predominantly neutral, when strong winds are present, sometimes stable/unstable but rarely very stable/unstable. Recalling that the wind speed increases

with inflection height the increase in neutral, stable and unstable conditions at higher inflections height simply reflects the stability distribution associated with the wind speeds at which they occur.

The Høvsøre results additionally indicate that stable conditions cause inflections at lower altitudes, similar to the results found offshore. If analyzing the height at which inflections occur during very stable conditions, the lowest height z=40m is the most common and the occurrence is found to monotonically decrease up to the highest inflection altitude of z=100m. This

could indicate a shallow surface layer during very stable conditions at Høvsøre and possibly being a part of the explanation for the progressive deviations in wind speed above 50-80m found at the Høvsøre site (Gryning et al., 2007).

### 4.4.3   The effect of stability on profile shapes

Describing the impact inflections have on the difference between expected and actual rotor equivalent wind speed is arguably the most effective way of communicating the implications of the results in this study. This impact was assessed through studying

the 1-inflection profile shapes with varying maximum height and atmospheric stability.

During neutral, stable and very stable conditions the inflections are found to have a more pronounced effect on the development of the vertical wind profile than during unstable/very unstable conditions (Fig. 10 and Fig. 11). Of these three, very stable conditions dominate offshore when inflections are present. Offshore, the very stably inflected profiles exhibit a decrease in wind speed over several height measurement above the maxima before the velocity profile reinstates an increasing-with-height

behaviour. When one additionally considers that MOST predicts largest wind speed gradients during very stable conditions, the very stable inflections seem to cause larger deviations from their predicted shape.

Very unstable conditions are contrary to the very stable profiles typically associated with low wind shear and a more uniform velocity profile both onshore and offshore, this is also reflected in the profile shapes seen in Fig. 10 and Fig. 11. Although the inflections in these profiles are not possible within the scope of MOST, a large amount of these inflections are too small to be



**Figure 12.** Wind direction of very stable and very unstable 1-inflection profiles at offshore sites. The wind roses are centered on the site location.

a likely cause of large deviations between the predicted and actual wind profile shape. Previous studies confirm that MOST is satisfactory in predicting the vertical wind profile during unstable conditions (Argyle and Watson, 2014)). It is however seen that some of the offshore 1-inflection profiles, especially at FINO1 and slightly at FINO2, show signs of a speed-up effect at the uppermost height. The most likely cause of this is thought to be the systematic speed-up effects which have been shown

5  to be present at FINO1, since analyzing the FINO2 mast-corrected velocity profiles removed the slight speed-up effects which are seen in the bottom middle plot of Fig. 10. (Westerhellweg et al., 2012). This explanation becomes more viable when one considers that FINO3 is equipped differently and does not have the 100m anemometer mounted on top of the mast, explaining why the FINO3 profiles show less signs of speed-up effects.

Onshore at Skipheia and Høvsøre the neutral profiles (middle plots, Fig. 11) may be of larger concern than the very stably

10  inflected profiles, since they occur more commonly and at higher mean wind speeds. The same can be said of the stable profiles,

**Figure 13.** Wind direction of very stable, neutral and very unstable 1-inflection profiles at onshore sites. The wind roses are centered on the site location.

whereas the unstable and very unstable profiles with inflections have low shear, small inflections and occur at low wind speeds, making them less relevant.

The amount of inflected profiles which may have a significant impact on the power production deficit was analyzed by considering profiles with a mean wind speed at 100m above 5 m/s, replicating a conservative cut-in wind speed at turbine hub

5 height. Assuming neutral, stable and very stable conditions to have the more severely inflected profiles, the following picture emerges: 16.8% of profiles are inflected under these conditions at FINO2, 10.0% at FINO1 and 3.7% at FINO3. Performing the same analysis at the coastal site Skipheia and the semi-coastal site Høvsøre, 10.43% of the inflected profiles at Skipheia fall within this category, and this sinks to 3.6% at Høvsøre. Offshore these profiles are predominantly occurring under very stable stratification, the coastal inflections are however more of a concern under weakly stable and near-neutral conditions. Evidently

10 there may be inflected profiles during unstable/very unstable conditions which are also sources of large deviations between





the predicted and actual 10-minute averaged wind profile, just as there are some very stably inflected profiles which are not as severe as they have been depicted here. Assuming these uncertainties are counter-balancing, these percentages depict the magnitude of the issues associated with inflected vertical wind profiles. There is however uncertainty related to the stability analysis and the exact number of inflected profiles may change with a different method of stability classification. Additionally there may be many profiles that deviate from the shape predicted by MOST without exhibiting an inflection. Future studies may therefore benefit from comparing predicted and measured wind speeds while simultaneously identifying inflected profiles. This can provide a more telling image on why each profile is deviating from its predicted shape.

The previous studies by Peña et al. (2008) and Sathe et al. (2012) both found that MO-theory over-predicted the offshore wind speed at higher elevations during stable conditions, which could be a result of the decreasing velocity found for several heights above the inflection point during stable conditions in the present study. Lange et al. (2004) found an over-prediction by MO-theory at z=50m, this could however be due to the height z=50m being within a stable capping inversion which has higher wind speeds and could in fact be a maximum in the vertical wind profile.

## 4.5 Wind direction of inflected profiles

The inflected profile categories as well as the height of profile inflections were checked for correlation to the direction of incoming wind. This was additionally performed after subdividing these profiles according to their atmospheric stability. The results show that the direction of a profile category is primarily dependant on the stability and wind speed under which it occurs. Therefore, the 1-inflection profiles are presented as an example here. The results were similar for higher order inflections since the direction is mainly decided by the stability regime associated with a profile category.

### 4.5.1 Offshore direction analysis

The wind rose for the very stable and very unstable 1-inflection profiles is shown in Fig 12. The wind rose is for illustration layered on top of a map centered on the site location, the locations can also be seen in Fig. 3 and Fig. 4. The neutral case was not shown for the offshore sites due to few occurrences.

At both FINO1 and FINO3 the inflected profiles during very stable conditions arrive from a sector of shorter fetch which was found to have a high occurrence of very stable winds at the respective sites. During very unstable conditions the inflected profiles arrive mostly from directional sectors with long fetch distances and the atmosphere is in these sectors predominantly very unstable. This result strongly indicates that the inflections during very stable conditions are occurring due to coastal effects and possibly a stable capping inversion.

At FINO2 the very unstable and very stable 1-inflection profiles do not show distinct sectors of higher occurrence. The atmospheric stability occurrence is more evenly distributed at FINO2, this causes a more even directional spread of both the very stable and very unstable inflections. The FINO2 site is located in the Baltic Sea and is more uniformly engulfed by land than FINO1 and FINO3.





### 4.5.2 Onshore direction analysis

The directional distribution of the 1-inflection profiles categorized during very stable, neutral and very unstable conditions at the onshore sites Skipheia and Høvsøre can be seen in Fig. 13.

Both the coastal site Skipheia and the semi-coastal site Høvsøre show a stability-dependency on the direction of incoming 5 1-inflection profiles. At both sites, the directions which are dominant during very stable, stable and very unstable conditions are sectors at each site where these conditions dominate. At both sites this results in very stable inflections arriving from onshore, whereas neutral and very unstable inflections are arriving from directions of short fetch and offshore incoming flow. The neutral inflections occur at the highest wind speeds, and are seen to arrive from offshore at both sites. These inflections may be related to shallow surface layers arriving from the sea.

### 10 5 Summary and conclusions

The occurrence of abnormal vertical wind profiles has been investigated to survey the applicability of Monin- Obukhov similarity theory in short term time-averaged vertical wind profiles. Through measurements the 10-minute averaged wind profile has been analyzed at seven sites up to a height of 100-140m depending on the site, where three of these are located offshore, two in coastal locations, one in a semi-coastal location and one is located inland surrounded by forest. Several years of data 15 was available at most sites, enabling a thorough comparison of how the occurrence of abnormalities changes with site location. The measured vertical wind profiles have been categorized in terms of the number of exhibited local maxima which are not possible within Monin-Obukhov similarity theory. With this identification method, the expected profile through MOST is the monotonically increasing profile with 0 inflections.

The results reveal that abnormal profiles are most common offshore, where data from the offshore masts FINO1 and FINO3 20 in the North Sea, and FINO2 in the Baltic Sea, reveals that inflected profiles occur in 65-75% of all analyzed 10-minute averages. The occurrence of abnormal profiles decreases as the roughness length increases and at the two onshore sites Valsneset and Skipheia in immediate vicinity to the Norwegian coast inflections are present in roughly 40% of the profiles. This percentage decreases to 16% only 1.7km inland at the Danish site Høvsøre, and is at its lowest occurrence of 12% at the inland Swedish forested location Ryningsnäs. The abnormal profiles were mainly in the form of 1 or 2 local maxima. Profiles with 25 more than 2 local maxima and cases with a reversed and monotonically decreasing profile were also identified, but these are rare events with less relevance in wind energy applications. They do however depict the limitations to MO-theory in describing the spectrum of occurring profile shapes.

The occurrence of abnormalities showed a strong correlation to wind speed and the thermal state of the atmosphere. Profiles with multiple inflections generally have a lower velocity, the spectrum of wind speeds where these profiles occur is however 30 large and many 1- and 2-inflection profiles occur at wind speeds relevant for wind energy applications. This is especially true at the offshore and coastal sites where the wind speed is generally higher.

Profiles inflected during positively buoyant very unstable conditions, i.e when there is strong turbulent mixing and low wind speed gradients, comprise the majority of abnormal profiles both onshore and offshore. The profiles inflected during conditions





of neutral and negative buoyancy, i.e neutral to very stable conditions, are less common, but due to their larger shear these profiles are proposed to be the source of largest deviations between the predicted and measured wind speeds. This issue was found to be most severe offshore in the Baltic Sea, where these inflections occur at turbine operating wind speeds in 16.8% of all profiles at FINO2 and provide a viable explanation for why previous studies have found that MOST incorrectly predicts the

vertical wind profile during stable conditions offshore (Sathe et al., 2012). While the Baltic Sea is a basin largely enclosed by land, vertical wind profiles severely inflected during very stable conditions were also found at FINO1 and FINO3 located 45km and 80km off- shore in the North Sea, and the wind direction of these profiles indicates an offshore internal boundary layer which may travel distances of more than 100km and still have large impacts on the vertical wind profile. At the coastal sites severe inflections are mainly linked with winds arriving from the sea during neutral conditions, with the coastal site Skipheia

exhibiting neutral/stable inflections above cut-in wind speed in 10.4% of all profiles. The occurrence of these profiles decreases rapidly further inland and is likely not visible more than a few kilometers onshore.

The results of this study do suggest an evident need of a more fulfilling vertical wind profile description, especially at coastal and offshore locations where high wind speeds and severe inflections occur simultaneously. A solution may be emerging through a unified vertical wind profile description which is valid through the entire atmospheric boundary layer and not

inhibited by surface-layer discontinuities. Such descriptions require knowledge of typical surface layer heights, and it is therefore important that future research continues the mapping of how the vertical wind profile develops under various surrounding conditions. This description is proposed to grow increasingly important as wind energy projects are expanding their reach to locations where little research has previously been conducted. While the study of inflected vertical wind profile has proved a simple and effective method for unambiguously categorizing abnormal vertical wind profiles, many additional profiles may be

incorrectly described by MOST without exhibiting such features. In order to better describe both the cause and implications of vertical wind profiles which deviate from the expected shape, future studies may benefit from a synthesized identification method of quantifying the error between the predicted and actual wind profiles while simultaneously describing the profile in terms of the number of local maxima.

*Data availability.* The Skipheia (Frøya) data is available online, see the link in Domagalski and Sætran (2019). Access to the remaining
datasets was granted as follows: FINO data may be granted through contact with personnel at BSH and through this available to download online. The Høvsøre data was made available through contact with Yoram Eisenberg of DTU Wind Energy. The Valsneset data was made available through contact with Lars Morten Bardal at NTNU. The Ryningsnäs data was made available through contact with Johan Arnqvist of Uppsala University.

*Author contributions.* The data analysis and manuscript preparation were both performed by Mathias Møller. Piotr Domagalski and Lars
Roar Sætran have mainly been involved in the project through technical discussions and manuscript revision.





*Competing interests.* The authors declare no competing interests.

*Acknowledgements.* For access to the Skipheia data, the authors would like to acknowledge the support of the European Commission in form of the MaRINET project (project ID: 262552) funded under FP7-INFRASTRUCTURES program. For access to the large FINO databases the authors would like to acknowledge BMWi (Bundesministerium fuer Wirtschaft und Energie, Federal Ministry for Economic Affairs and Energy ) and the PTJ (Projekttraeger Juelich, project executing organization), and also Anne-Christin Shulz from BSH for answering questions regarding the datasets. For access to the Høvsøre data, the authors would like to thank the DTU Wind Energy Energy group, and especially Yoram Eisenberg for providing the data. The authors would also like to thank Johan Arnqvist at University of Uppsala for providing access to the Ryningsnäs data, and Lars Morten Bardal of NTNU for access to the Valsneset data.





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
