# Peer review of "Comparing Abnormalities in Onshore and Offshore Vertical Wind Profiles"

_Wind Energy Science, 2019_

## Referee Comment (RC1) · Anonymous Referee #1 · 19 Aug 2019

REVIEW:

Comparing Abnormalities in Onshore and Offshore Vertical Wind Profiles. WES-2019-40

The proposed article discusses abnormalities in the vertical wind speed profile from several years os observation at six sites in Northern Europe. Abnormalities are detected by local maximum in the vertical wind speed profile that cannot occur in MOST. The number and height of observed abnormalities are statistically correlated to the mean wind speed, to thermal stability. A comparison onshore / offshore is made. Conclusions are a frequent of appearance of theses abnormalities from 65% offshore to 40% of the time onshore.

General comments: The article is well in the scope of WES and presents interesting original results. It has an extensive and well organized literature study. The main issue is that conclusions are very much weakened by having, in my opinion, only one onshore site (that has a very "extreme" roughness case) and by the missing analysis of the intensity of the maximum detected: in the present work, the most tiny deviation from MOST, that will have no effect on wind turbine operation, is accounted at an equivalent level as a large fluctuation that will certainly affect significantly the power production/loading...

The manuscript is a little long, the writing should be more concise

Major remarks:

1- MOST 1.1 At several locations in the manuscript, you mention the hypothesis needed for applying MOST without really introducing them. Page 2 lines 9-10: Insist on the hypothesis used to build MOST (explicitly mention the surface layer) and give an approximate value of the region where MOST is valid. 1.2 When discussing abnormalities compared to MOST, do you consider cases where MOST is applicable (all MOST hypothesis are fulfilled) but observations differ from theory or cases where MOST cannot be applied? In the first case the theory is threatened and in the second not as you analyze cases where MOST cannot be applied. This is fundamentally different, please comment on that. Ultimately, it doesn't affect the interest of the work in studying VWP.

2- Methodology In the method of determining local maximum, the number of maximum and their height are recorded. However, (at least up to section 4.4.2), the smallest local maximum has the same importance as a very large deviation. The former would have an imperceptible impact on wind turbine performance/production/loads but the latter, would possibly have a large one. You mention at the end, section 4.4.3, that most of the maximum found are in unstable conditions where local maximum are very weak, that means with very limited effect for wind turbines. Would a methodology accounting

for the intensity of the peaks (maybe eliminating the smallest peaks?) lead to the same results and conclusions? Ultimately, what is the effect on your conclusion of 65%-75% (p.28 l.20) of profiles inflected? Among this ratio, how many maximums are really affecting WT operation?

3- Onshore/Coastal/onshore sites 3.1 The article claims to analyze both onshore and offshore sites. In reality, according to me, mainly offshore sites are discussed as only one site is really onshore (Ryningsnas). Hovsore, Skipheia and Valsneset are clearly much coastal as observed in fig 13. Ryningsnas is in a forest, witch is a bit "extreme" in terms of surface roughness. The discussion on the effect of surface roughness on abnormal events would really be improved by the analysis of several "really" onshore sites with more moderate roughness. 3.2 What is the sense of analyzing coastal sites as a whole? It may make more sense to divide coastal sites in function of the wind direction, a offshore fetch and an onshore fetch? This is partially confirmed by fig13. 3.3 Can you make appear the offshore/coastal/onshore classification in on of the tables detailing the sites?

4- Stability 4.1 Stability bins seems to be the same at all sites (p.14 l.1), is it realistic for both offshore and forest sites? See for example: Sanz-Rodrigo et al. Journal of Physics: Conference Series 625 (18 juin 2015): 012044. Dupont et al. Agricultural and Forest Meteorology 157 (15 mai 2012): 11‑29 4.2 How much is the sensibility of the choice of the stability classes (p.27 l.4)? Can you give an order of magnitude.

Minor remarks:

Are there other tools to detect abnormalities (deviations from MOST)?

The mast speed up effect description (p.18 l.14-16: and p.25 l.2-8) is very important, it has to be in the site description part.

In the description of the measurement equipment, more information are needed on the LiDAR data: time/space resolution, volume probed. . . The LiDAR data may be affected

by longitudinal and vertical space-average that may smooth out small maximum (p18 l.15-16)?

p19, l29 → p20, l2: I don't understand this part. What do you mean by "spectrum of velocity"? Do you mean turbulent spectrum? Something else? Why don't you show them? Additionally, I don't understand what you get from these "spectrum"... Also mentioned p.28 l.29

p.22 l.16-17: "it was found to be due..." rather approximative statement. You need more proofs to say that. Better say you enlighten a correlation...

p.20 l.19-21: A shallow surface layer is a possible explanation. Could it be estimated from sonic anemometer profile to verify your hypothesis?

Technical corrections:

Revise the use of abbreviations for Sec. Fig. Tab. ... Use Figure sub-numbering when more than 2 figure (a,b,c,d...)

p.2 l.21-22: unclear sentence

p.3 l.7: define IBL the first time you use it p.3 l.11: "short-lived phenomena" → I guess you speak about space rather than time, reword to make it clearer.

p.4 l.18: why a new paragraph here? p.4 l.19: this sentence is a bit "lost" here... p.4 l.26-27: please rephrase

Fig. 2 is cited much later in the text, please move it at the right location.

Tab. 1 and 2 can be merged in one. Remove all information not necessary for the present paper (was pressure used? Humidity?)

Section 3.1, please move all references to the way you got the data to the acknowledgements.

p.14 l.14: define MABL

Fig 5: The sorting seems to be linked to Z0, a better choice of colors would make the reading easier. For example changing the bars filling as function of on-shore/coastal/offshore.

p.15 l.8: "observed" may be better appropriated than "displayed"

p.18 l.3:5 and figure 6: why not plotting occurrences in a scatter plot (such as fig7 "middle") that would help comparison. And all sites in the same plot.

p.16 l.16: double "that" to remove.

Fig 7: the central and right plots can be merged, one of your goals is to underline the difference onshore/offshore, potting in the same graph will enhance comparison.

p.19 l.16-17: this sentence has already been said p17 l18-20

p19 l23: change "These profiles..." by "The latter profiles..."

p.24 l.14: "Recalling that the reference wind speed at 100m increases..." p.24 l.14: add a coma "...height, the increase"

p.28 l.12: remove "Through"

Fig12: please switch the two figure on the right to make the figure consistent.

---

## Referee Comment (RC2) · Anonymous Referee #2 · 18 Oct 2019

The paper presents a useful and thorough analysis of the shape of wind speed profiles at onshore, coastal offshore and offshore sites. This is quite timely as people question the applicability of MOST even in flat locations such as offshore. The quality of results presented is clear and analyses anomaliies in wind profiles as a function of several relevant parameters.

My main concern is the sigificance of the maxima in the profiles studied. As pointed out by the authors, under unstable conditions, the wind shear is much reduced and thus fairly small wind speed changes can create maxima. The same is true when comparing offshore with onshore, especially forested sites. The higher roughness length will give higher shear and thus reduce the influence of wind speed fluctuations in terms of giving rise to maxima. So the results presented seem more about the variation in maxima as

a fucntion of average wind shear. The approach could be improved if a maximum is only recorded as such if a threshold is reached. This could be in terms of a fixed wind speed value or some sort of 95% exceedance, for example. The authors need to show that the 'kinks' in the profile are more than just an artefact resulting from differences in turbulent fluctuations at different heights which are more significant under lower shear conditions. If a thresholding approach were done, this would make the analysis much stronger and highlight specific phenomena, e.g. low level jets, which are likely to cause a deviation form MOST.

Specific comments: Figure 2 does not seem to be referenced or described in the text.

---

## Author Comment (AC1) · 4 Nov 2019

We thank Anonymous Referee 1 for the time and comments. In the following, we will do our best to engage the comments and propose improvements for the final manuscript.

Anonymous Referee #1 general comments: The proposed article discusses abnormalities in the vertical wind speed profile from several years old observations at six sites in Northern Europe. Abnormalities are detected by local maximum in the vertical wind speed profile that cannot occur in MOST. The number and height of observed abnormalities are statistically correlated to the mean wind speed to thermal stability. A comparison onshore / offshore is made. Conclusions are a frequent of appearance of theses abnormalities from 65% offshore to 40% of the time onshore. The article is

well in the scope of WES and presents interesting original results. It has an extensive and well organized literature study. The main issue is that conclusions are very much weakened by having, in my opinion, only one onshore site (that has a very "extreme" roughness case) and by the missing analysis of the intensity of the maximum detected: in the present work, the most tiny deviation from MOST, that will have no effect on wind turbine operation, is accounted at an equivalent level as a large fluctuation that will certainly affect significantly the power production/loading. . . The manuscript is a little long, the writing should be more concise

The authors' reply to Reviewer 1 general comment: We gladly find the General comments positive. The main issues, the sites availability, possible extension of analysis into deviation intensity analysis are in details answered in details below. We agree that the text should be more concise and will be shortened it in the corrected version.

Anonymous Referee #1 major remarks: Anonymous Referee major remark 1: MOST 1.1 At several locations in the manuscript, you mention the hypothesis needed for applying MOST without really introducing them. Page 2 lines 9-10: Insist on the hypothesis used to build MOST (explicitly mention the surface layer) and give an approximate value of the region where MOST is valid. 1.2 When discussing abnormalities compared to MOST, do you consider cases where MOST is applicable (all MOST hypothesis are fulfilled) but observations differ from theory or cases where MOST cannot be applied? In the first case the theory is threatened and in the second not as you analyze cases where MOST cannot be applied. This is fundamentally different, please comment on that. Ultimately, it doesn't affect the interest of the work in studying VWP.

The authors' reply to Anonymous Referee major remark 1: MOST 1.1 The authors appreciate the comment, and agree that the assumptions of MOST should be outlined more clearly. The following is suggested to be added after lines 9-10 on page 2: "The logarithmic law describes the vertical development of velocity in the surface layer which is typically only the shallowest 10% of the atmospheric boundary layer. The depth of the surface layer where MO-theory is valid varies with the state of the atmosphere, from

only a few meters during very stable stratification to several hundred meters during conditions of vigorous turbulent mixing. ". 1.2 The authors agree that this point should be clarified, and suggest an edit of the last sentence of the introduction to: "Based on this the applicability of the commonly used vertical wind profiles may be evaluated, and the need for more accurate vertical wind profile descriptions can be discussed. It is emphasized that the scope of the paper focuses on the applicability of the theory commonly employed today, and not the validity of the theory itself"

Anonymous Referee major remark 2: Methodology In the method of determining local maximum, the number of maximum and their height are recorded. However, (at least up to section 4.4.2), the smallest local maximum has the same importance as a very large deviation. The former would have an imperceptible impact on wind turbine performance/production/loads but the latter, would possibly have a large one. You mention at the end, section 4.4.3, that most of the maximum found are in unstable conditions where local maximum are very weak, that means with very limited effect for wind turbines. Would a methodology accounting for the intensity of the peaks (maybe eliminating the smallest peaks?) lead to the same results and conclusions? Ultimately, what is the effect on your conclusion of 65%- 75% (p.28 l.20) of profiles inflected? Among this ratio, how many maximums are really affecting WT operation?

The authors' reply to Anonymous Referee major remark 2: Methodology The authors certainly agree that such an analysis is an interesting pursuit. An initial analysis of this issue is given in section 4.4.2, and the percentages of profiles "severely affected" are given in the conclusion. The authors have concluded to perform an additional analysis where the inflections are grouped in bins according to their severity, which will be added as a separate subsection. We do however feel that enforcing a severity threshold for the entire analysis limits the wider scope and relevance of the work which in its current form is not only limited to wind turbine engineering. We do agree that a focus on inflection severity is highly relevant for wind engineering and highly encourage this issue to be the focus of future work. For this reason we have, as mentioned, decided to

have a stronger focus on this issue in the present manuscript through adding another subsection to discuss the topic more in depth.

Anonymous Referee major remark 3: Onshore/Coastal/offshore sites 3.1 The article claims to analyze both onshore and offshore sites. In reality, according to me, mainly offshore sites are discussed as only one site is really onshore (Ryningsnas). Høvsøre, Skipheia and Valsneset are clearly much coastal as observed in fig 13. Ryningsnas is in a forest, witch is a bit "extreme" in terms of surface roughness. The discussion on the effect of surface roughness on abnormal events would really be improved by the analysis of several "really" onshore sites with more moderate roughness.

3.2 What is the sense of analyzing coastal sites as a whole? It may make more sense to divide coastal sites in function of the wind direction, a offshore fetch and an on-shore fetch? This is partially confirmed by fig13. 3.3 Can you make appear the off-shore/coastal/onshore classification in one of the tables detailing the sites?

The authors' reply to Anonymous Referee major remark 3: Onshore/Coastal/Offshore sites 3.1 The authors agree that the discussion may have been raised by having more traditional onshore sites. The data sites were however limited to the sites where we had available data. We are proud to have gathered and analyzed a large amount of data from several sites and we are grateful for access granted to the data. We do however agree with Anonymous Referee 1 and regret not to have access to data from sites at a traditional onshore location, that would enhance the analysis.

3.2 Regarding the possibility of splitting the coastal sites into onshore and offshore sectors, this is the scope of another paper currently in press (https://iopscience.iop.org/issue/1742-6596/1356/1). This was not done in the current manuscript since the change in occurrence of inflections from onshore to coastal to offshore was the main scope. The authors believe the dependence of infection occurrence on site location was clearly portrayed without splitting the coastal sites into onshore/offshore sectors, it was therefore avoided.

[Figure]

Anonymous Referee major remark 4: Stability 4.1 Stability bins seems to be the same at all sites (p.14 l.1), is it realistic for both offshore and forest sites? See for example: Sanz-Rodrigo et al. Journal of Physics: Conference Series 625 (18 juin 2015): 012044. Dupont et al. Agricultural and Forest Meteorology 157 (15 mai 2012): 11âAËŸ S29 4.2 How much is the sensibility of the choice of the stability classes (p.27 l.4)? Can you give an order of magnitude.

The authors' reply to Anonymous Referee major remark 3: Onshore/Coastal/Offshore sites 4.1 The authors agree that the stability bins ideally should not be the same for onshore/offshore locations, as presented in suggested works, which investigate this issue in depth. However that would make the direct comparison between the sites impossible. Sanz-Rodrigo et al. (2015) states that his seven classes are 'somehow ambiguous' and in the conclusions of his work we can read that 'it is convenient to adopt certain conventions when it comes to measuring and defining stabilities". That is exactly what we did: for simplification and comparison ease we followed 5 bins classification, as we found it is used for offshore sites as well, for example in the following works:.

Barthelmie, R. J., Churchfield, M. J., Moriarty, P. J., Lundquist, J. K., Oxley, G. S.,Hahn, S., Pryor, S. C., 2015, "The role of atmospheric stability/turbulence on wakes at the Egmond aan Zee offshore wind farm," J. Phys. Conf. Ser.,625(1), p. 012002.

Barthelmie, R. J. (1999). The effects of atmospheric stability on coastal wind climates. Meteorological Applications, 6(1), 39-47.

Motta, M., Barthelmie, R. J., & Vølund, P. (2005). The influence of non‐logarithmic wind speed profiles on potential power output at Danish offshore sites. Wind Energy: An International Journal for Progress and Applications in Wind Power Conversion Technology, 8(2), 219-236.

Alblas, L., Bierbooms, W., Veldkamp, D., 2014, "Power output of offshore windfarms in relation to atmospheric stability," J. Phys. Conf. Ser.555(1), p. 01200

4.2 In authors opinion, changing the stability bins classification between 5/7/9 bins will not change the main findings of the manuscript, however we are grateful for pointing this issue and we will possibly switch to more than 5 bins classification scheme in our future work. Anonymous Referee #1 minor remarks: Anonymous Referee minor remark 1: Are there other tools to detect abnormalities (deviations from MOST)?

The authors' reply to Anonymous Referee minor remark 1: Yes, there are many ways to detect deviations from MOST. As briefly mentioned in the Summary and conclusions, future studies may benefit from using several methods for identifying abnormalities. One method could be measuring the deviation from the vertical wind profile predicted by MOST (i.e deviation from a log-law profile) for each 10-minute profile. Anonymous referee minor remark 2: The mast speed up effect description (p.18 l.14-16: and p.25 l.2-8) is very important, it has to be in the site description part.

The authors' reply to Anonymous Referee minor remark 2: The authors agree that this is critical and will move this to the description in the revised manuscript.

Anonymous referee minor remark 3: In the description of the measurement equipment, more information is needed on the LiDAR data: time/space resolution, volume probed. . . The LiDAR data may be affected r by longitudinal and vertical space-average that may smooth out small maximum (p18 l.15-16)? The authors' reply to Anonymous Referee minor remark 3: The authors agree that additional information regarding the LiDAR measurement should be given, this will be provided in the revised manuscript. To our knowledge no evidence of smoothing of maximum has been found.

Anonymous referee minor remark 4: p19, l29 → p20, l2: I don't understand this part. What do you mean by "spectrum of velocity"? Do you mean turbulent spectrum? Something else? Why don't you show them? Additionally, I don't understand what you get from these "spectrum". . .

The authors' reply to Anonymous Referee minor remark 4: By spectrum the authors simply meant within the range of observed velocities. This will be clarified in the revised

manuscript.

Anonymous referee minor remark 5: Also mentioned p.28 l.29 p.22 l.16-17: "it was found to be due..." rather approximative statement. You need more proofs to say that. Better say you enlighten a correlation. . . The authors' reply to Anonymous Referee minor remark 5: The authors agree and will change this in the revised manuscript.

Anonymous referee minor remark 6: p.22 l.19-21: A shallow surface layer is a possible explanation. Could it be estimated from sonic anemometer profile to verify your hypothesis?

The authors' reply to Anonymous Referee minor remark 6: This could be a possible explanation. The sonic measurement data does not allow for this to be verified due to data availability issues, but it will be added as a possible explanation.

Anonymous Referee #1 technical corrections: The authors have found it most efficient to only comment on the technical corrections if we are not in agreement with the technical corrections. Otherwise, the suggested corrections listed below will be implemented in the revised manuscript.

Anonymous referee technical correction 1: Revise the use of abbreviations for Sec. Fig. Tab. . ..

Anonymous referee technical correction 2: Use Figure sub-numbering when more than 2 figure (a,b,c,d...)

Anonymous referee technical correction 3: p.2 l.21-22: unclear sentence

Anonymous referee technical correction 4: p.3 l.7: define IBL the first time you use it

Anonymous referee technical correction 5: p.3 l.11: "short-lived phenomena" → I guess you speak about space rather than time, reword to make it clearer.

Anonymous referee technical correction 6: p.4 l.18: why a new paragraph here?

Anonymous referee technical correction 7: p.4 l.19: this sentence is a bit "lost" here. . .

Anonymous referee technical correction 8: p.4 l.26-27: please rephrase Fig. 2 is cited much later in the text, please move it at the right location.

Anonymous referee technical correction 9: Tab. 1 and 2 can be merged in one. Remove all information not necessary for the present paper (was pressure used? Humidity?)

The authors' reply to Anonymous Referee technical correction 9: Pressure and humidity were used in the stability calculations for calculating the potential virtual temperature.

Anonymous referee technical correction 10: Section 3.1, please move all references to the way you got the data to the acknowledgements.

Anonymous referee technical correction 11: p.14 l.14: define MABL

Anonymous referee technical correction 12: Fig 5: The sorting seems to be linked to Z0, a better choice of colors would make the reading easier. For example changing the bars filling as function of onshore/coastal/offshore.

Anonymous referee technical correction 13: p.15 l.8: "observed" may be better appropriated than "displayed"

Anonymous referee technical correction 14: p.18 l.3:5 and figure 6: why not plotting occurrences in a scatter plot (such as fig7 "middle") that would help comparison. And all sites in the same plot.

Anonymous referee technical correction 15: p.16 l.16: double "that" to remove.

Anonymous referee technical correction 16: Fig 7: the central and right plots can be merged, one of your goals is to underline the difference onshore/offshore, potting in the same graph will enhance comparison.

The authors' reply to Anonymous Referee technical correction 16: This was done intentionally as gathering all lines in one plot created too many plotted lines resulting in an unclear plot. The range on the y-axis is however the same for both plots.

Anonymous referee technical correction 17: p.19 l.16-17: this sentence has already been said

Anonymous referee technical correction 18: p17 l18-20 p19 l23: change "These profiles..." by "The latter profiles..."

Anonymous referee technical correction 19: p.24 l.14: "Recalling that the reference wind speed at 100m increases..."

Anonymous referee technical correction 20: p.24 l.14: add a coma "...height, the increase"

Anonymous referee technical correction 21: p.28 l.12: remove "Through"

Anonymous referee technical correction 22: Fig12: please switch the two figure on the right to make the figure consistent.

Please also note the supplement to this comment:
https://www.wind-energ-sci-discuss.net/wes-2019-40/wes-2019-40-AC1-supplement.pdf

---

## Author Response (AR1)

**Authors' point-by-point response to comments on "Comparing Abnormalities in Onshore and Offshore Vertical Wind Profiles" WES-2019- 40 by Mathias Møller, Piotr Domagalski, and Lars Roar Sætran**

**Mathias Møller, Piotr Domagalski, and Lars Roar Sætran**

We thank both Anonymous Referee 1 (AR1) and Anonymous Referee 2 (AR2) for the comments provided. In the following, we will provide a point-by-point response to all referee comments and the corresponding manuscript changes. When highlighting the changes, the page and line numbers referred to are all in the marked-up manuscript, this is done to facilitate simple change tracking.

**Anonymous referee 1 general comments:**

The proposed article discusses abnormalities in the vertical wind speed profile from several years old observations at six sites in Northern Europe. Abnormalities are detected by local maximum in the vertical wind speed profile that cannot occur in MOST. The number and height of observed abnormalities are statistically correlated to the mean wind speed to thermal stability. A comparison onshore / offshore is made. Conclusions are a frequent of appearance of theses abnormalities from 65% offshore to 40% of the time onshore. The article is well in the scope of WES and presents interesting original results. It has an extensive and well organized literature study. The main issue is that conclusions are very much weakened by having, in my opinion, only one onshore site (that has a very "extreme" roughness case) and by the missing analysis of the intensity of the maximum detected: in the present work, the most tiny deviation from MOST, that will have no effect on wind turbine operation, is accounted at an equivalent level as a large fluctuation that will certainly affect significantly the power production/loading. . . The manuscript is a little long, the writing should be more concise

**The authors' reply to AR1 general comment:**
We gladly find the AR1 general comments positive. The main issues, the sites availability, possible extension of analysis into deviation intensity analysis are in details answered below. We agree that the text should be more concise and will be shortened it in the corrected version.

**Changes due to AR1 general comments:**
The manuscript has in several locations been shortened due to the general comment of the manuscript being a little long. These changes are highlighted in the marked-up manuscript tracking the changes made from the original discussion paper.

**Anonymous Referee 1 major remarks:**

**Anonymous Referee 1 major remark 1: MOST**
*1.1* At several locations in the manuscript, you mention the hypothesis needed for applying MOST

without really introducing them. Page 2 lines 9-10: Insist on the hypothesis used to build MOST (explicitly mention the surface layer) and give an approximate value of the region where MOST is valid.

*1.2* When discussing abnormalities compared to MOST, do you consider cases where MOST is applicable (all MOST hypothesis are fulfilled) but observations differ from theory or cases where MOST cannot be applied? In the first case the theory is threatened and in the second not as you analyze cases where MOST cannot be applied. This is fundamentally different, please comment on that. Ultimately, it doesn't affect the interest of the work in studying VWP.

**The authors' reply to AR1 major remark 1:  MOST**
**1.1** The authors appreciate the comment and agree that the assumptions of MOST should be outlined more clearly.
**1.2** The authors agree that this point should be clarified and suggest an edit of the last sentence of the introduction.

**Changes due to AR1 major remark 1: MOST**
**1.1:** On page 2 line 13 the following has been added:
"*The logarithmic law describes the vertical development of velocity in the surface layer which is typically only the shallowest 10% of the atmospheric boundary layer. The depth of the surface layer where MO-theory is valid varies with the state of the atmosphere, from only a few meters during very stable stratification to several hundred meters during conditions of vigorous turbulent mixing.*"

**1.2:** On page 4 line 29 the sentence has been changed to the following:
"*Based on this the applicability of the commonly used vertical wind profiles may be evaluated, and the need for more accurate vertical wind profile descriptions can be discussed. It is emphasized that the scope of the paper focuses on the applicability of the theory commonly employed today, and not the validity of the theory itself.*"

*Anonymous referee 1 major remark 2: Methodology*
In the method of determining local maximum, the number of maximum and their height are recorded. However, (at least up to section 4.4.2), the smallest local maximum has the same importance as a very large deviation. The former would have an imperceptible impact on wind turbine performance/production/loads but the latter, would possibly have a large one. You mention at the end, section 4.4.3, that most of the maximum found are in unstable conditions where local maximum are very weak, that means with very limited effect for wind turbines. Would a methodology accounting for the intensity of the peaks (maybe eliminating the smallest peaks?) lead to the same results and conclusions? Ultimately, what is the effect on your conclusion of 65%- 75% (p.28 l.20) of profiles inflected? Among this ratio, how many maximums are really affecting WT operation?

**The authors' reply to AR1 major remark 2: Methodology**
The authors certainly agree that such an analysis is an interesting pursuit. An initial analysis of this issue is given in section 4.4.2, and the percentages of profiles "severely affected" are given in the conclusion. The authors have concluded to perform an additional analysis where the inflections are analyzed in terms of their severity, which will be added as a separate subsection. We do however feel that enforcing a severity threshold for the entire analysis limits the wider scope and relevance of the work which in its current form is not only limited to wind turbine engineering. We do agree that a

focus on inflection severity is highly relevant for wind engineering and highly encourage this issue to be the focus of future work. For this reason, we have, as mentioned, decided to have a stronger focus on this issue in the present manuscript through adding another subsection to discuss the topic more in depth.

**Changes due to AR1 major remark 2:**
The subsection *4.5 Inflection severity* has been added on page 27 line 4, along with Figure 12:

"

*4.5 Inflection severity*

*The inflection severity was analyzed in order to support the findings of the profile shapes presented in Section 4.4.3. The severity of an inflection was defined as the difference in speed at the point of the maximum and the wind speed at the point where the velocity profile retains its positive shear above the inflection ($\Delta U_{flic} = U_{inflection} - U_{min\ above\ inflection}$ [m/s]). The results shown in Fig. 12 reveal that $\Delta U_{flic}$ is typically small during unstable conditions which matches the flat velocity profiles described in Section 4.4.3. This may indicate that the very unstable inflections are to a larger degree caused by small arbitrary variations in the vertical wind speed which may be caused by turbulent fluctuations. The very stable 1-inflection profiles are contrarily seen to have a much larger inflection severity at all sites, again showing that the very stable inflections are more critical in wind energy applications. By assuming an inflection is severe if $\Delta U_{flic} \leq 0.5$ the results show that, depending on the site, 9-25% of all 1-inflection profiles are categorised as severe. For the very unstable 1-inflection profiles only 3-14% of profiles are severe, and for very stable 1-inflection profiles as many as 35-48% of the inflected profiles are severely inflected. Although slight variations between sites are found the results clearly illustrate that once an inflected profile has been identified, the likelihood of a severe inflection is much higher during stable conditions.*

"

*Anonymous referee 1 major remark 3: Onshore/Coastal/offshore sites*
*3.1* The article claims to analyze both onshore and offshore sites. In reality, according to me, mainly offshore sites are discussed as only one site is really onshore (Ryningsnas). Høvsøre, Skipheia and Valsneset are clearly much coastal as observed in fig 13. Ryningsnas is in a forest, witch is a bit "extreme" in terms of surface roughness. The discussion on the effect of surface roughness on abnormal events would really be improved by the analysis of several "really" onshore sites with more moderate roughness.
*3.2* What is the sense of analyzing coastal sites as a whole? It may make more sense to divide coastal sites in function of the wind direction, a offshore fetch and an onshore fetch? This is partially confirmed by fig13. 3.3 Can you make appear the offshore/coastal/onshore classification in one of the tables detailing the sites?

**The authors' reply to AR1 major remark 3: Onshore/Coastal/Offshore sites**
**3.1** The authors agree that the discussion may have been raised by having more traditional onshore sites. The data sites were however limited to the sites where we had available data. We are proud to have gathered and analyzed a large amount of data from several sites and we are grateful for access granted to the data. We do however agree with Anonymous Referee 1 and regret not to have access

to data from sites at a traditional onshore location, that would enhance the analysis.

**3.2** Regarding the possibility of splitting the coastal sites into onshore and offshore sectors, this is the scope of another paper currently in press (https://iopscience.iop.org/issue/1742-6596/1356/1). This was not done in the current manuscript since the change in occurrence of inflections from onshore to coastal to offshore was the main scope. The authors believe the dependence of infection occurrence on site location was clearly portrayed without splitting the coastal sites into onshore/offshore sectors, it was therefore avoided.

_Anonymous referee 1 major remark 4: Stability_

_4.1_ Stability bins seems to be the same at all sites (p.14 l.1), is it realistic for both offshore and forest sites? See for example: Sanz-Rodrigo et al. Journal of Physics: Conference Series 625 (18 juin 2015): 012044. Dupont et al. Agricultural and Forest Meteorology 157 (15 mai 2012): 11âAˇ S29

_4.2_ How much is the sensibility of the choice of the stability classes (p.27 l.4)? Can you give an order of magnitude.

**The authors' reply to AR1 major remark 3: Onshore/Coastal/Offshore sites**

**4.1** The authors agree that the stability bins ideally should not be the same for onshore/offshore locations, as presented in suggested works below which investigate this issue in depth. That would however make the direct comparison between the sites challenging. Sanz-Rodrigo et al. (2015) states that his seven classes are 'somehow ambiguous' and in the conclusions of his work we can read that 'it is convenient to adopt certain conventions when it comes to measuring and defining stabilities". That is exactly what has been done int the present work: for simplification and comparison ease we followed 5 bins classification, as we found it is used for offshore sites as well, for example in the following works:.

_Barthelmie, R. J., Churchfield, M. J., Moriarty, P. J., Lundquist, J. K., Oxley, G. S.,Hahn, S., Pryor, S. C., 2015, "The role of atmospheric stability/turbulence on wakes at the Egmond aan Zee offshore wind farm," J. Phys. Conf. Ser.,625(1), p. 012002._

_Barthelmie, R. J. (1999). The effects of atmospheric stability on coastal wind climates. Meteorological Applications, 6(1), 39-47._

_Motta, M., Barthelmie, R. J., & Vølund, P. (2005). The influence of non-logarithmic wind speed profiles on potential power output at Danish offshore sites. Wind Energy: An International Journal for Progress and Applications in Wind Power Conversion Technology, 8(2), 219-236._

_Alblas, L., Bierbooms, W., Veldkamp, D., 2014, "Power output of offshore windfarms in relation to atmospheric stability," J. Phys. Conf. Ser.555(1), p. 01200_

**4.2** In authors opinion, changing the stability bins classification between 5/7/9 bins will not change the main findings of the manuscript, however we are grateful for pointing this issue and we will possibly switch to more than 5 bins classification scheme in our future work.

**_Anonymous Referee #1 minor remarks:_**

*Anonymous referee 1 minor remark 1:*
Are there other tools to detect abnormalities (deviations from MOST)?

**The authors' reply to AR1 minor remark 1:**
Yes, there are many ways to detect deviations from MOST. As briefly mentioned in the Summary and conclusions, future studies may benefit from using several methods for identifying abnormalities. One method could be measuring the deviation from the vertical wind profile predicted by MOST (i.e deviation from a log-law profile) for each 10-minute profile.

*Anonymous referee 1 minor remark 2:*
The mast speed up effect description (p.18 l.14-16: and p.25 l.2-8) is very important, it has to be in the site description part.

**The authors' reply to AR1 minor remark 2:**
The authors agree that this is critical and will move this to the description in the revised manuscript.

**Changes due to AR1 minor remark 2:**
Sentence *"The FINO1 mast has been shown to be prone to mast speed-up effects (Westerhellweg et al., 2012), the handling of this issue is discussed in section 3.2.1."* has been moved to Section 3.1.2, page 10 Line 15.

*Anonymous referee 1 minor remark 3:*
In the description of the measurement equipment, more information is needed on the LiDAR data: time/space resolution, volume probed. . . The LiDAR data may be affected r by longitudinal and vertical space-average that may smooth out small maximum (p18 l.15-16)?

**The authors' reply to AR1 minor remark 3:**
The authors agree that additional information regarding the LiDAR measurement should be given, this is provided in the revised manuscript. To our knowledge no evidence of smoothing of maximum has been found.

**Changes due to AR1 minor remark 3:**
The following has been added on page 12 line 20:

"…*using the Leosphere WINDCUBE V2. The LiDAR has a measurement frequency5of 1 Hz, a velocity accuracy of 0.1 m/s and a directional accuraccy of 2 $^{o}$*"

*Anonymous referee 1 minor remark 4:*
p19, l29 → p20, l2: I don't understand this part. What do you mean by "spectrum of velocity"? Do you mean turbulent spectrum? Something else? Why don't you show them? Additionally, I don't understand what you get from these "spectrum". . .

**The authors' reply to AR1 minor remark 4:**
By spectrum the authors simply meant within the range of observed velocities. This is clarified in the revised manuscript.

**Changes due to AR1 minor remark 4:**
The sentence *"To investigate the impact of these abnormal profiles, the spectrum of velocity within*

*which the profiles occur was checked*" on page 20 line 18 was removed since it was deemed confusing.

*Anonymous referee 1 minor remark 5:*
Also mentioned p.28 l.29 p.22 l.16-17: "it was found to be due..." rather approximative statement. You need more proofs to say that. Better say you enlighten a correlation. . .

**The authors' reply to AR1 minor remark 5:**
The authors agree that such sentences should be rephrased.

**Changes due to AR1 minor remark 5:**
On page 22 line 12 changed to "*The results suggest this is*", page 29 line 1 changed to "*The results indicate this issue is*".

*Anonymous referee 1 minor remark 6:*
p.22 l.19-21: A shallow surface layer is a possible explanation. Could it be estimated from sonic anemometer profile to verify your hypothesis?

**The authors' reply to AR1 minor remark 6:**
This could be a possible explanation. The sonic measurement data does not allow for this to be verified due to data availability issues, but it will be clarified as a possible explanation.

**Changes due to AR1 minor remark 6:**
On page 23 line 8 the sentence "*No apparent reason for this increase was found, the possibility of these inflections heights being coupled with longer fetch distances was checked but did not yield an explanation, neither did a cross-correlation with wind speed*" was removed and the two paragraphs were merged together.

**Anonymous Referee #1 technical corrections:**

The authors have found it most efficient to only comment on the technical corrections if we are not in agreement with the technical corrections. Otherwise, the suggested corrections listed below will be implemented in the revised manuscript.

*Anonymous referee 1 technical correction 1:*
Revise the use of abbreviations for Sec. Fig. Tab. .

**Authors response to AR1 technical correction 1:**
According to the manuscript preparation guidelines Fig, Sec and Tab should be used in mid-sentence, and the whole should be spelled out if it initiates a new sentence. The authors have followed this principle.

*Anonymous referee 1 technical correction 2:*
 Use Figure sub-numbering when more than 2 figure (a,b,c,d...)

**Authors response to AR1 technical correction 2:**
According to manuscript preparation guidelines no additional packaged should be added and only

one figure file should be provided per panel. The current implementation is in line with these instructions.

*Anonymous referee 1 technical correction 3:*
p.2 l.21-22: unclear sentence

**Authors response to AR1 technical correction 3:**
The authors do not find this sentence particularly unclear and have disregarded this comment.

*Anonymous referee 1 technical correction 4:*
p.3 l.7: define IBL the first time you use it

**Authors response to AR1 technical correction 4:**
The authors have now done this on page 1 line 4.

*Anonymous referee 1 technical correction 5:*
p.3 l.11: "short-lived phenomena" → I guess you speak about space rather than time, reword to make it clearer.

**Authors response to AR1 technical correction 5:**
This is correct, the word "*spatial*" has been added on page 3 line 14.

*Anonymous referee 1 technical correction 6:*
p.4 l.18: why a new paragraph here?

**Authors response to AR1 technical correction 6:**
The authors agree that a new paragraph here is not necessary.

*Anonymous referee 1 technical correction 7:*
p.4 l.19: this sentence is a bit "lost" here. . .

**Authors response to AR1 technical correction 7:**
The authors do not find anything particularly peculiar with this sentence, it is therefore not changed.

*Anonymous referee 1 technical correction 8:*
p.4 l.26-27: please rephrase Fig. 2 is cited much later in the text, please move it at the right location.

**Authors response to AR1 technical correction 8:**
The authors believe the Figure should be where it is in the text and have added "*and a random selection of profiles from one measurement site is shown in Fig. 2.*" on page 6 line 2.

*Anonymous referee 1 technical correction 9:*
Tab. 1 and 2 can be merged in one. Remove all information not necessary for the present paper (was pressure used? Humidity?)

**The authors' reply to AR1 technical correction 9:**
Pressure and humidity were used in the stability calculations for calculating the potential virtual temperature. The authors have not found a good way to merge the tables and this has therefore not been done.

*Anonymous referee 1 technical correction 10:*
Section 3.1, please move all references to the way you got the data to the acknowledgements.

**Authors response to AR1 technical correction 10:**
The authors agree, and this has been moved to the acknowledgements.

*Anonymous referee 1 technical correction 11:*
p.14 l.14: define MABL

**Authors response to AR1 technical correction 11:**
The authors agree, and this has been defined on page 14 line 16.

*Anonymous referee 1 technical correction 12:*
Fig 5: The sorting seems to be linked to Z0, a better choice of colors would make the reading easier. For example changing the bars filling as function of onshore/coastal/offshore.

**Authors response to AR1 technical correction 12:**
The sorting is done onshore ->offshore, so it is linked to $z_0$. The current color scheme has been chosen after testing many variations and was found to be the optimal choice available.

*Anonymous referee 1 technical correction 13:*
p.15 l.8: "observed" may be better appropriated than "displayed"

**Authors response to AR1 technical correction 13:**
The word displayed is changed to "*observed*" on page 15 line 8.

*Anonymous referee 1 technical correction 14:*
p.18 l.3:5 and figure 6: why not plotting occurrences in a scatter plot (such as fig7 "middle") that would help comparison. And all sites in the same plot.

**Authors response to AR1 technical correction 14:**
The authors have tried this previously but have found that the current visualization is more clarifying. Having all sites in the same plot gave too much information and was visually unpleasing.

*Anonymous referee 1 technical correction 15:*
p.16 l.16: double "that" to remove.

**Authors response to AR1 technical correction 15:**
The authors have removed "*that*" on page 17 line 6.

*Anonymous referee 1 technical correction 16:*
Fig 7: the central and right plots can be merged, one of your goals is to underline the difference onshore/offshore, potting in the same graph will enhance comparison.

**The authors' reply to AR1 technical correction 16:**
This was done intentionally as gathering all lines in one plot created too many plotted lines resulting in an unclear plot. The range on the y-axis is however the same for both plots.

*Anonymous referee 1 technical correction 17:*
p.19 l.16-17: this sentence has already been said

**Authors response to AR1 technical correction 17:**
The authors agree and the sentence on page 20 line 2 has been removed.

*Anonymous referee 1 technical correction 18:*
p17 l18-20 p19 l23: change "These profiles..." by "The latter profiles..."

**Authors response to AR1 technical correction 18:**
The authors changed this formulation on page 20 line 13 to "*Such profiles are however*". The second usage of the formulation was a part of the removed sentence on page 20 line 18.

*Anonymous referee 1 technical correction 19:*
p.24 l.14: "Recalling that the reference wind speed at 100m increases..."

**Authors response to AR1 technical correction 19:**
The authors agree that this formulation is clarifying, and "*at z=100m*" has been added on page 24 line 14.

*Anonymous referee 1 technical correction 20:*
p.24 l.14: add a coma "...height, the increase"

**Authors response to AR1 technical correction 20:**
A comma has been added on page 24 line 15

*Anonymous referee 1 technical correction 21:*
p.28 l.12: remove "Through"

**Authors response to AR1 technical correction 21:**
Through has been removed on page 28 line 10.

*Anonymous referee 1 technical correction 22:*
Fig12: please switch the two figure on the right to make the figure consistent.

**Authors response to AR1 technical correction 22:**
The two figures in the now Fig. 13 have been switched on page 27.
* * *
**Anonymous referee 2 general comment:**

The paper presents a useful and thorough analysis of the shape of wind speed profiles at onshore, coastal offshore and offshore sites. This is quite timely as people question the applicability of MOST even in flat locations such as offshore. The quality of results presented is clear and analyses anomaliies in wind profiles as a function of several relevant parameters.

My main concern is the sigificance of the maxima in the profiles studied. As pointed out by the authors, under unstable conditions, the wind shear is much reduced and thus fairly small wind speed changes can create maxima. The same is true when comparing offshore with onshore, especially forested sites. The higher roughness length will give higher shear and thus reduce the influence of

wind speed fluctuations in terms of giving rise to maxima. So the results presented seem more about the variation in maxima as a fucntion of average wind shear. The approach could be improved if a maximum is only recorded as such if a threshold is reached. This could be in terms of a fixed wind speed value or some sort of 95% exceedance, for example. The authors need to show that the 'kinks' in the profile are more than just an artefact resulting from differences in turbulent fluctations at different heights which are more significant under lower shear conditions. If a thresholding approach were done, this would make the analysis much stronger and highlight specific phenomena, e.g. low level jets, which are likely to cause a deviation form MOST.

**The authors' reply to AR2 general comment**
This comment is largely the same as what was highlighted by AR1 in major remark 2. The response to this comment is therefore repeated here: The authors certainly agree that such an analysis is an interesting pursuit. An initial analysis of this issue is given in section 4.4.2, and the percentages of profiles "severely affected" are given in the conclusion. The authors have concluded to perform an additional analysis where the inflections are analyzed in terms of their severity, which will be added as a separate subsection. We do however feel that enforcing a severity threshold for the entire analysis limits the wider scope and relevance of the work which in its current form is not only limited to wind turbine engineering. We do agree that a focus on inflection severity is highly relevant for wind engineering and highly encourage this issue to be the focus of future work. For this reason, we have, as mentioned, decided to have a stronger focus on this issue in the present manuscript through adding another subsection to discuss the topic more in depth.

*Anonymous referee 2 specific comment:*
Figure 2 does not seem to be referenced or described in the text.

**Authors response to AR1 technical correction 14:**
The figure is referenced in several locations, for example in the revised manuscript on page 6 line 2 (in latexdiff document).

[revised manuscript text omitted]

~~The wind profile shape (i.e shear) was also studied, these results are discussed in Section 4.4 since wind shear is intrinsically coupled to the thermal state of the atmosphere. It should also be mentioned that there were certain deviations found from the aforementioned general findings. FINO2 is for example not seen to show only a very slight decrease in wind speed at z=100m when the number of inflections increases incrementally from 1 to 3. Comparing the wind speed at the lowest height reveals that the wind speed remains very constant between 1 and 3 inflections, a result unique for this site.~~

[revised manuscript text omitted]

---

## Author Response (AR2)

**Authors' point-by-point response to Revised Submission comments on "Comparing Abnormalities in Onshore and Offshore Vertical Wind Profiles" WES-2019- 40 by Mathias Møller, Piotr Domagalski, and Lars Roar Sætran**

**Mathias Møller, Piotr Domagalski, and Lars Roar Sætran**

We thank both Anonymous Referee 1 (AR1) and Anonymous Referee 3 (AR3) for valuable comments helping to improve the manuscript. In the following, we will provide a point-by-point response to Anonymous Referee #3's comments and the corresponding manuscript changes. When highlighting the changes, the page and line numbers referred to are all in the marked-up manuscript, this is done to facilitate simple change tracking.

**_Anonymous referee 3 general comments:_**

The representativeness of the onshore sites. Three sites are coastal and one is located in a forested region. It could have been interesting to include an onshore flat and open site in the analysis, which could have been a kind of reference regarding the applicability of MOST.

The methodology which has been applied, which consists in detecting local maximums or minimums in the wind speed profiles. These maximums or minimums are characterized by their number, without any consideration of their magnitude. Very weak inflections in the wind speed profile are not very significant (taking into account the uncertainties and the representativeness of the measurements) and will have no impact on wind turbines, and more generally won't be relevant in wind engineering. A filtering could have been applied on the data set in order to retain only significant inflections in the analysis. The new paragraph on "inflection severity" answers partially to this weakness by providing the occurrence probability as a function of the inflection magnitude (for 1-inflection profiles only). But it shows that imposing a minimum of 0.2 m/s for this magnitude would remove a large part (up to 70-75% for some sites) of the detected inflections. I would suggest to take that into account in the conclusions and in the abstract. For example, I think that the proportion of inflected profiles which is mentioned for offshore sites (65-75%) is a bit misleading, due to the high occurrence of unstable cases for these sites, for which the inflections are most of the time not really significant and have probably no practical impact.

However this paper can still be considered as a useful contribution to the characterization of the wind vertical profiles.

**The authors' reply to AR1 general comment:**

The authors thank AR3 for the generally positive comments. Regarding the issue of the weakly inflected profiles, the authors are in full agreement that the weakly inflected profiles have little impact on wind turbines. In the previous revision of the manuscript the authors argued that employing a severity threshold for all inflections would limit the wider scope and relevance of the

work which in its current form is not only limited to wind turbine engineering. Covering this issue did lead to the additional paragraph "4.5 Inflection Severity" in revision of the manuscript. This is also pointed out by AR3. The authors also agree that the leading results in the abstract and conclusion should give an additional focus to the occurrence of severely inflected profiles. The second part of the sentence on page 1 line 12 is therefore changed to:

 "....causing up to 17% severely inflected abnormal profiles at the most exposed offshore site, which decreases as the location transitions from offshore to coastal to further inland, and is lowest at the forested site."

**Anonymous Referee 3 specific comments:**

**Anonymous Referee 3 major remark 1: Abstract**

I don't think we can say that "the occurrence of local maxima scales inversely to the roughness length". This occurrence results from a combination of different physical processes among which other parameters (especially the stability) play an important role. Page 15 line 4 the authors mentioned that "the onshore occurrence of abnormal profiles is found to scale inversely with the distance to shore", which is a different conclusion.

**The authors' reply to AR3 major remark 1:**
The authors fully agree that the occurrence of local maxima is dependent on more than only the roughness length, the wording "scales inversely" was therefore chosen. For clarity, the phrasing is on page 1 line 12 changed to:

"The results reveal that the occurrence of local maxima is higher at sites of low surface roughness and a high prevalence of unstable atmospheric conditions."

**Anonymous Referee 3 major remark 2: Paragraph 3.2:**
I am surprised that no direction sector was excluded from the analysis for the 3 FINO masts. For example, it is explained in paragraph 3.1.4 that for the FINO3 mast, only the anemometer installed in the direction 345° was used for consistency between vertical levels. Therefore a sector filtering should have been applied, especially regarding the very weak inflections of wind speed which are considered in the analysis.

**The authors' reply to AR3 major remark 2:**
In general the authors share the impression that analysis of FINO masts data and in fact all wind mast data, requires careful thought on how to deal with mast distortion effects. The authors do however believe this issue has already been discussed in section 3.2.1 "filtering mast distortion" of the current manuscript. We quote from the section:

"The data from the FINO sites includes mast corrected wind speeds based on a uniform ambient flow correction (UAM) algorithm (Westerhellweg et al., 2012), mast corrected wind speeds were however only available at all heights at the FINO2 site. The analysis in this study was conducted on both the mast-corrected and non-corrected wind speeds at FINO2, as well as both including and excluding the mast-distorted sectors at all FINO sites. The results were found to be similar in all cases, thus no mast

distortion filtering was employed. The same conclusion was drawn by Kettle (2014) when studying local maxima at FINO1"

*Anonymous Referee 3 major remark 3: Paragraph 3.3*

Please mention the vertical levels that have been used to evaluate the stability. As mentioned by the authors, the distribution of stability was found by Argyle and Watson (2014) as very sensitive to the measuring heights used. In their paper, the occurrence of very unstable conditions was found to vary between about 38% to 70% depending on these heights. Thus it is difficult to state that there is a reasonable agreement between the two papers on this occurrence.

**The authors' reply to AR3 major remark 3:**
The authors agree that this information should be present and suggest the following edit on page 13 line 26:

"At the offshore sites, the following heights were used in the stability calculation: 50m and 70m at FINO1 and FINO2, and 30m and 55m at FINO3. At the onshore sites the following height was used in addition to the surface temperature: 40m at Skipheia, and 100m at Høvsøre.  These heights were chosen due to high data availability, and where possible the main correlations between atmospheric stability and wind profile inflections have been cross-checked using different stability measurement heights. The cross-reference was found to strengthen the main findings related to atmospheric stability. There is nonetheless …...."

*Anonymous Referee 3 major remark 4: Page 19 lines 15-17:*

The sentence is not clear: if I understand correctly, the stability is not associated to the inflection height itself but to the time stamp corresponding to the occurrence of this inflection.

**The authors' reply to AR3 major remark 4:**
The authors are in full agreement with AR3, it is of course the time stamp corresponding to the occurrence of the inflection which is relevant. For clarity the authors suggest the following change to the manuscript on page 19 line 10:

"The occurrence at each height is however also found to be strongly coupled with the atmospheric stability at the time of profile inflection. This is discussed in Section 4.4.2 which addresses the changes in stability distribution when inflections occur at varying heights."

*Anonymous Referee 3 major remark 5: Page 20 line 11*

The height at which the wind speed is considered should be mentioned

**The authors' reply to AR3 major remark 5:**
It is in fact the mean wind profile which is considered and not the wind speed at a specific height. This way the comparison of data from different sites, with different wind speed measurements levels, is possible. This is clarified in the manuscript by the following suggested edit on page 10 line 12:

"The inflections during very unstable conditions show significantly larger variation in  the *magnitude of the mean wind speed profile* with a changing maximum height, which can be seen at FINO3 (the

site with the largest variation) by the order from left to right of the profiles in the top right plot of Fig. 10 versus the bottom right plot of Fig 10."

*Anonymous Referee 3 major remark 6: Page 21 lines 10-13*

The sentence is not clear, please rephrase it.

**The authors' reply to AR3 major remark 6:**
The authors address the comment by suggesting the following edit to the manuscript on page 22 line 4:

"Considering the offshore sites FINO1 and FINO2 first, the occurrence of a very stable atmosphere is higher when the maximum occurs at a lower altitude (≈ 30% and ≈ 50% respectively), however when a maximum occurs at a higher altitude the occurrence of stable conditions is lower and instead very unstable conditions becomes increasingly dominant."

*Anonymous Referee 3 major remark 7: Page 24 line 11*

The decrease in very unstable conditions appears clearly for Skipheia, but is not so clear for Hovsore.

**The authors' reply to AR3 major remark 7:**
The authors agree with this remark and suggest the following edit on page 23 line 8:

"Onshore (Skipheia and Høvsøre) there is an increase in neutral, slightly stable and slightly unstable conditions as the inflection rises. …."

**Anonymous Referee #3 technical corrections:**

The authors have found it most efficient to only comment on the technical corrections if we are not in agreement with the technical corrections. Otherwise, the suggested corrections listed below will be implemented in the revised manuscript.

*Anonymous referee 3 technical correction 1:*

p. 5 l. 12: I would suggest to define Nmax in a general way without reference to a Python function.

**The authors reply to AR3 technical correction 1:**
The authors thank AR3 for the comment, and agree that Python functions should not be used. The definition is however also given in the manuscript in terms of the mathematical floor operators. Including the floor()-definition was done to enlighten readers unfamiliar with this mathematical operator. The authors therefore believe that the definition will be more clearly communicated if it stands as it is at the current state.

*Anonymous referee 3 technical correction 2:*

p. 11 l. 23: "an" should be replaced by "and"

*Anonymous referee 3 technical correction 3:*

p. 12 l. 21: "accuraccy should be replaced by "accuracy"

*Anonymous referee 3 technical correction 4:*

p. 17 l. 3: "the results of FINO1" should be replaced by "the results of FINO1 are"

*Anonymous referee 3 technical correction 5:*

p. 18 l. 1: "confirm that offshore sites to a larger degree experience local maximum in the wind profile than onshore sites" should be replaced by "confirm that offshore sites experience local maximum in the wind profile to a larger degree than onshore sites"

*Anonymous referee 3 technical correction 6:*

p. 19 l. 4: "the cause of this is result" should be replaced by "the cause of this result"

*Anonymous referee 3 technical correction 7:*

p. 21 l. 10: "specifically the the" should be replaced by "specifically the"

*Anonymous referee 3 technical correction 8:*

p. 24 in the figure 10 caption : "not shown" instead of "not show"

*Anonymous referee 3 technical correction 9:*

p. 26 l. 7: "over several height measurement" should be replaced by "over several measurement heights"

*Anonymous referee 3 technical correction 10:*

p. 27 l. 5: "the difference in speed at" should be replaced by "the difference between the wind speed at"

*Anonymous referee 3 technical correction 11:*

P. 28 l. 7: If the paragraph "Wind direction of inflected profiles" is removed, please remove the title of this paragraph as well.

**The authors' response to AR3 technical correction 11:**

The section has not been removed, the authors have however noticed that this section was missing in the manuscript outlining the differences between the initial and revised submission. For this the authors sincerely apologize, the error will be corrected for the final submission.

*Anonymous referee 3 technical correction 12:*

P. 28 l. 11: "has" should be replaced by "have"

*Anonymous referee 3 technical correction 13:*

p. 28 l. 13: "was" should be replaced by "were"

**The authors' response to AR3 technical correction 13:**

The authors thank AR3 for the comment but do believe "was" has been used correctly.

*Anonymous referee 3 technical correction 14:*

p.28 l. 27: I suggest to replace "spectrum" by "range"
* * *

[revised manuscript text omitted]